

# Surface snow bromide and nitrate at Eureka, Canada in early spring and implications for polar boundary layer chemistry

Xin Yang[1], Kimberly Strong[2], Alison S. Criscitiello[3], Marta Santos-Garcia[1*], Kristof Bognar[2**], Xiaoyi Zhao[4], Pierre Fogal[2], Kaley A. Walker[2], Sara M., Morris[5], and Peter Effertz[6,7]

[1]British Antarctic Survey, Natural Environment Research Council, Cambridge, UK
[2]Department of Physics, University of Toronto, Toronto, ON, Canada
[3]Department of Earth and Atmospheric Sciences, University of Alberta, Edmonton, Alberta, Canada
[4]Air Quality Research Division, Environment and Climate Change Canada, Toronto, ON, Canada
[5]NOAA Earth System Research Laboratories, Physical Sciences Laboratory, Boulder, CO, USA
[6]Cooperative Institute for Research in Environmental Science - CU Boulder, Boulder, CO, USA
[7]NOAA Earth System Research Laboratories, Global Monitoring Laboratory, Boulder, CO, USA

[*]Now at School of Geosciences, University of Edinburgh, Edinburgh, UK

[**]Now at 3v Geomatics Inc., Vancouver, BC, Canada

*Correspondence to*: Xin Yang (xinyang55@bas.ac.uk)

**Abstract.** This study explores the role of snowpack in polar boundary layer chemistry, especially as a direct source of reactive bromine ($BrO_X=BrO+Br$) and nitrogen ($NO_X=NO+NO_2$) in the Arctic springtime. Surface snow samples were collected daily from a Canadian high Arctic location at Eureka, Nunavut (80°N, 86°W) from the end of February to the end of March in 2018 and 2019. The snow was sampled at several sites representing distinct environments: sea ice, inland close to sea level, and a hilltop ~600 m above sea level (asl). At the inland sites, surface snow salinity has a double-peak distribution with the first and

lowest peak at 0.001-0.002 practical salinity unit (psu), which corresponds to the precipitation effect, and the second peak at 0.01–0.04 psu, which is likely related to the salt accumulation effect (due to loss of water vapour by sublimation). Snow salinity on sea ice has a triple-peak distribution; its first and second peaks overlap with the inland peaks, and the third peak at 0.2–0.4 psu is likely due to the sea water effect (due to upward migration of brine on sea ice). At all sites, snow sodium and chloride concentrations increase by almost 10-fold from the top 0.2 cm to ~1.5 cm in depth. Surface snow bromide at sea level is

significantly enriched, indicating a net sink of atmospheric bromine. Moreover, surface snow bromide at sea level has an increasing trend over the measurement time period, with mean slopes of 0.024 in the 0-0.2 cm layer and 0.016 μM d[-1] in the 0.2–0.5 cm layer. Surface snow nitrate at sea level also shows a significant increasing trend, with mean slopes of 0.27, 0.20, and 0.07 μM d[-1] in the top 0.2 cm, 0.2–0.5 cm, and 0.5–1.5 cm layers, respectively. Using these trends, an integrated net deposition flux of bromide of $1.01\times10^7$ molecules cm[-2] s[-1] and an integrated net deposition flux of nitrate of $2.6\times10^8$ molecules

cm[-2] s[-1] were derived. In addition, nitrate and bromide in the morning samples are significantly higher than the afternoon samples, indicating a strong photochemistry effect. However, the mean bromide loss rate (0.027-0.040 μM) is smaller than the




nitrate loss rate (0.23-0.362 μM) by an order of magnitude, implying the reactive bromine emission flux from snowpack is significantly smaller than the reactive nitrogen emission flux, which is consistent with the large difference between their derived net deposition fluxes. After considering the photochemical loss effect, the corrected bromide deposition flux at sea

level is $2.73 \times 10^7$ molecules cm$^{-2}$ s$^{-1}$; for nitrate, the corrected deposition flux is $5.98 \times 10^8$ molecules cm$^{-2}$ s$^{-1}$. In addition, the surface snow nitrate and bromide at inland sites were found to be significantly correlated (R=0.48-0.76), and the [NO$_3^-$]/[Br$^-$] ratio of 4-7 indicates a possible acceleration effect of reactive bromine in atmospheric NO$_X$-to-nitrate conversion. This is the first time such an effect has been seen in snow chemistry data obtained with a sampling frequency as short as one day.

## 1 Introduction

Reactive bromine (BrO$_X$=BrO+Br) and reactive nitrogen (NO$_X$=NO+NO$_2$) are two important families in atmospheric chemistry, both of which play a critical role in determining the oxidising capacity of the polar boundary layer (Morin et al., 2008). However, the processes involved in the sources, sinks, and recycling of reactive bromine and nitrogen in the air-snow-sea ice system are not fully understood (Abbatt et al., 2012) or parameterised, which prevents quantification of their effects and the ability to make robust predictions for the changing climate using numerical chemical models.

Reactive nitrogen-rich air observed in the Arctic troposphere is mainly anthropogenic and subject to long-range transport (Dickerson, 1985). During winter, gaseous nitric acid (HNO$_3$) or particulate bond nitrate (p-NO$_3$) is removed from the air via dry and wet deposition. HNO$_3$ and p-NO$_3$ mainly dissolve to form nitrate (NO$_3^-$) upon contact with the snow cover (Diehl et al., 1995; Abbatt, 1997). Nitrate that accumulates in snowpack can release gaseous NO$_X$ and HONO in spring via photolysis (Dubowski et al., 2001; Honrath et al., 2002), with the processes controlled by many factors including

meteorological parameters and chemical, optical, and physical snow properties. These include photolabile NO$_3^-$ concentrations, the amount of light-absorbing impurities, the temperature-dependent quantum yields of NO$_3^-$ photolysis, and the timing of precipitation (Beine et al., 2003; Frey et al., 2013; Chan et al., 2015; Zatko et al., 2016; Winton et al., 2020). The measured snow-NO$_X$ emission fluxes in polar regions vary from site to site, ranging from near zero to $>1.0 \times 10^9$ molecules cm$^{-2}$ s$^{-1}$ (Jones et al., 2001; Zhou et al., 2001; Honrath et al., 2002; Beine et al., 2002; 2003; Oncley et

al., 2004; Frey et al., 2013; Chan et al., 2018). A direct measurement of nitrate dry deposition flux was made by Björkman et al. (2013) in Svalbard using a tray sampling approach. They reported a total flux of $10.27 \pm 3.84$ mg m$^{-2}$ (September 2009 to May 2010) which is roughly equivalent to a mean flux of $4 \times 10^8$ molecules cm$^{-2}$ s$^{-1}$. In addition, at Svalbard, precipitation dominates nitrate supply to snow, with dry deposited HNO$_3$ only accounting for 10-14% of total nitrate (Beine et al., 2003; Björkman et al., 2013).

Observations show that sea-ice regions have the highest tropospheric bromine oxide (BrO) loading on Earth (Wagner and Platt, 1998). BrO enhancements are normally observed in the polar boundary layer during springtime and are referred to as "bromine explosion" events (BEEs). It is well known that saline substrates are the eventual source of reactive bromine (Wagner and Platt, 1998; Oum et al., 1998; Simpson et al., 2007a). Salts may be supplied to the snow surface by upward



migration from sea ice, by frost flowers being wind-blown to the snow surface, or by wind-transported aerosols generated by

sea spray (Domine et al., 2004). However, the dominant source bromine and the underlying processes involved remain unclear, with more than half a dozen different candidates proposed. These include frost flowers (Kaleschke et al., 2004; Piot and von Glasow, 2008), first-year sea ice surface (Simpson et al., 2005; 2007b), open leads/polynyas (e.g., Peterson et al., 2016; Kirpes et al., 2019; Criscitiello et al., 2021), snowpack on tundra (Pratt et al., 2013), snowpack on sea ice (Custard et al., 2017; Peterson et al., 2019), snowpack on ice sheets (Thomas et al., 2011), and sea salt aerosols from blowing snow (Yang et al.,

2008; 2010; 2019, 2020; Frey et al., 2020; Huang et al., 2020). Significant progress has been made in recent decades, with data showing that frost flowers and open leads are only of minor or local importance (Domine et al., 2005; Obbard et al., 2009; Huang et al., 2020). In addition, the proposed stratospheric BrO intrusion (Salawitch et al., 2010) has also been found to be less important than previously thought (Theys et al., 2011). Currently, the major debate surrounds the relative importance of the two remaining candidates – snowpack and blowing snow (e.g., Bognar et al., 2020; Marelle et al., 2021; Swanson et al.,

75 2022).

Reactive bromine can directly cause polar boundary layer ozone depletion events (ODEs), whereby near-surface ozone concentrations in spring drop below 10 ppbv (part per billion by volume), reaching close to 0 ppbv in some cases (Bottenheim et al., 1986; Barrie et al., 1988; Tarasick and Bottenheim, 2002; Jacobi et al., 2012). In addition, $BrO_X$ can affect reactive nitrogen (Morin et al., 2008) and hydroxyl radicals ($HO_X = OH + HO_2$) (Bloss et al., 2007, 2010; Brough et al., 2019)

as well as elemental mercury oxidation (e.g., Holmes et al., 2006; Parella et al., 2012; Angot et al., 2016; Xu et al., 2016; Wang et al., 2019) and dimethyl sulphide oxidation (Hoffmann et al., 2016).

It is well-known that $BrO_X$ can directly react with $NO_X$ via the following reactions:

$$BrO(g) + NO_2(g) \rightarrow BrONO_2(g) \tag{R1}$$
$$BrONO_2(g) + H_2O(aq) \rightarrow HNO_3(g) + HOBr(g) \tag{R2}$$

Thus, the presence of $BrO_X$ may accelerate the conversion from NOx to nitrate and influence the atmospheric nitrogen budget. Previous modelling work has estimated that bromine chemistry can cause $NO_X$ reductions of 60-80% at high latitudes in spring (Yang et al., 2005).

The emission fluxes of reactive bromine from blowing snow are all based on parameterisation in models (Yang et al., 2008; 2010, 2020; Huang et al., 2020; Swanson et al., 2020; Marelle et al., 2021). There are currently no direct measurements

of bromine emission flux from blowing snow. Regarding snowpack bromine emission, a direct gradient measurement of $Br_2$ and BrCl above a patch of snowpack was made near Utqiaġvik, Alaska (Custard et al., 2017), who reported emission fluxes of $0.7$–$12 \times 10^8$ molecules $cm^{-2}$ $s^{-1}$. However, their emission fluxes were based on a field dataset obtained over only a few days. Model emission schemes estimated reactive bromine emission fluxes of $9.0 \times 10^7$ to $2.7 \times 10^9$ molecules $cm^{-2}$ $s^{-1}$, and the emission flux is highly dependent on the parameters applied (Lehrer et al., 2004; Poit et al., 2009; Toyota et al., 2014; Falk

and Sinnhuber, 2018; Marelle et al., 2021). The removal of inorganic bromine species (such as HBr, HOBr, $Br_2$, BrCl, $BrONO_2$ and BrO) from the atmosphere via wet and dry depositions is mainly calculated by models (e.g., Yang et al., 2005; 2010; Parella et al., 2012; Legrand et al., 2016), so far there has not deposition flux reported.





Both nitrate and bromide undergo post-depositional processing within the snowpack (i.e. photochemistry), and the observationally derived flux represents the net direction of emission and deposition. A net source of $Br_2$ and BrCl was measured

over snowpack which was enriched in bromide (Custard et al., 2017), this is likely due to the fact that deposition and emission are two different processes and they may occur at different time, in different depth and with different rate. For example, the deposited bromide and nitrate may be largely confined to the top few cm layer, while photochemistry may occur only in daytime and across a deep depth (depending on the e-folding depth (Domine et al., 2008)).

Different methods have been used to derive the flux of deposited ions to snow (Cadle et al., 1991; Beine et al., 2003;

Macdonald et al., 2017). For instance, Björkman et al. (2013) applied three different methods to derive nitrate dry deposition flux at Svalbard: tray sampling, glacial sampling, and modelling. Macdonald et al. (2017) derived major ions (including nitrate) deposition fluxes at Alert, Nunavut, from freshly fallen snow samples collected on average every four days. However, they could not derive bromide deposition flux, which could be due to the efficient post-depositional loss of bromide, given the sampling interval varying from 1 to 19 days. In this study we apply a methodology similar to, but slightly different from, that

used by Macdonald et al. (2017). For example, we deliberately increase temporal sampling resolution to ~24 hours and collect snow samples directly from the snowpack surface using a vertical resolution of 2-3 mm. This vertical resolution will enable us to collect flesh falling snow from trace precipitation (an amount of precipitation greater than zero, but too small to be measured by standard methods of measurement). Because of mixing of surface snow particles due to wind, samples collected in the skin layer are not solely from snow recently fallen in the past 24 hours (with exceptions in very calm conditions), rather

they represent a mixture of various snow particles. Thus, ions measured in the surface layer are not only due to deposition in the past 24 hours deposition, but also to deposition in previous days. Therefore, by looking into the average change of ions within a time scale of 24 hours, we could be able to derive a mean net deposition flux of ions such as nitrate and bromide. To that end, we collected the top 1.5 cm of snow in three sub-layers: 0-0.2, 0.2-0.5, and 0.5-1.5 cm at several sampling sites (including onshore and offshore sites as well as on the top of a hill) in the high Arctic at Eureka (80°N, 86°W), Nunavut,

Canada (Figure 1) on a daily basis during early spring in 2018 and 2019. The aim of this study is to derive a net deposition flux of bromide and nitrate to surface snow and then to infer the role that snowpack plays as a direct source of reactive bromine and nitrogen. Methods and datasets are described in Section 2. The results are reported in Section 3. Discussions and atmospheric implications of this study are in Section 4, with conclusions given in Section 5.

## 2 Methods and datasets

### 2.1 Sampling site and local meteorology

Eureka is one of the coldest and driest places in the Canadian Arctic, with average air temperature of -37°C and precipitation of ~2 mm in March. Surface inversions are frequently observed in winter-spring (~84% of the time), and boundary layer height is in the range of 400-800 m (Bradley et al., 1992). Due to the local geography and cold weather, sea ice near the Eureka Weather Station (EWS) is thick (e.g., >1.5 m in early spring) and stable. Satellite-based sea ice data show that there



are no clearly identifiable leads or open waters within 600-800 km to the north and west of Eureka in early spring (Bognar et al., 2020). Therefore, the impact of local open leads is negligible. In addition, modelling work shows that this area is only weakly influenced by open ocean sea spray (Rhodes et al., 2017), thus open-ocean sourced bromine influence is of secondary importance (Yang et al., 2020). Under calm weather conditions, the atmospheric boundary layer at Eureka is generally shallow and stratified. Thus, the measurements made at the Polar Environment Atmospheric Research Laboratory (PEARL) Ridge Laboratory, located on the top of a hill (610 m asl) (Figure 1) are mainly representative of the free tropospheric influence; however, under unstable condition such as cyclones, The PEARL Ridge Lab is within the extended boundary layer. In early spring, the UV index changes dramatically from very low levels at the end of February to higher levels at the end of March (Figure S1), mainly due to the rapid increase in daily solar elevation angles after polar sunrise on February 21.

Sea water starts to freeze in late September at Eureka, with snow accumulated in the following months (before December). Therefore snowpack depth does not change much after December, which is consistent with the result of an Arctic snow depth survey by Warren et al. (1999). On sea ice, snowpack depth near EWS is 10-30 cm, while snow depth inland varies from only a few cm at convex locations to more than half a meter at concave locations. The type of sea ice in the Slidre Fiord is mainly one-year ice. However, a large iceberg was grounded in the fiord since Summer 2018, which seems significantly affect 2019 snow salinity and ionic concentrations on sea ice (see section 3).

## 2.2 Snow sampling

As can be seen from Figure 1, several sampling sites were located between EWS and the PEARL Ridge Lab. The two major sampling sites at sea level are ~5 km to the west of EWS: one on sea ice (named "Sea ice," ~100 m offshore) and one onshore (named "Onshore," ~50 m inland). There are two additional inland sites (also close to sea level) just behind EWS: the PEARL "0PAL" (Zero Altitude PEARL Auxiliary Laboratory) site and the "Creek" site which are close together and ~1000 m from the sea ice. The PEARL Ridge Lab (hereafter, referred to as PEARL) is another major sampling site, which is ~15 km to the west of EWS on top of a hill. In addition, a few snow samples were collected from the Eureka airport (~70 m asl, ~3 km to the east of EWS) and on the sea ice in front of EWS; however, these samples were not for ionic analysis due to local contamination concerns.

There are two types of surface snow observed at Eureka. One consists of fluffy mobile snow particles, loosely connected and white in colour. They mainly cover the top 0.5 cm of snow, and are a mixture of recent falling snow, drifting snow, and deposited ice crystals. On slightly raised surfaces that face the predominant winds, there is a wind-crust layer that is light brown in colour and hard to break, representing aged snow. In 2018, these two types of surface snow were deliberately collected for salinity analysis. All samples were collected using their sampling tubes to simply scratch them from the surface, roughly at a depth of 0.3–0.5 cm.

A small patch of snow (about 1 m by 2 m) was identified at each major sampling site (Sea ice, Onshore and PEARL) for daily snow sampling. In 2019, surface snow was collected using a small shovel with a funnel. Since March 4 daily snow samples were collected from three sub-layers (0-0.2, 0.2-0.5 and 0.5-1.5 cm). To investigate local geographic variation, a few



snow samples were randomly collected across a distance of 1-2 m at each sampling site (from two snow layers: 0-0.5 and 0.5-1.5 cm between February 26 and March 3). On March 4 and 5, validation samples were collected during a precipitation event
from three snow layers (0-0.2, 0.2-0.5 and 0.5-1.5 cm) at the 0PAL, Onshore and Sea ice sites.

In addition to surface snow, airborne snow samples were collected on a daily basis using a mounted tray outside. For example, one tray was mounted outside the 0PAL building (~1 m above the ground), and another one was mounted on the roof of the PEARL Ridge Lab (~1.5 m above the roof and ~11 m above the ground). In windy conditions, most of the samples collected by trays consist of blowing snow particles. In calm conditions, trace samples from deposited ice crystals and growing
hoar frost at the edge of the tray can be collected. During precipitation events, freshly falling snow can be sampled.

For logistical reasons, the time of day for surface snow sampling could not be fixed.  Samples were normally collected either in the morning (9:30-11:00 AM local time) or in the afternoon (2:30-5:00 PM local time). This enables the samples to be used to investigate the photochemistry effect.  Since March 15, 2019, the majority of samples from Sea ice and Onshore were collected in the afternoon.

Column snow samples were collected (at a vertical resolution of 1-3 cm) from a few sampling sites at irregular intervals, but mainly during March 4–12 in both 2018 and 2019. Ionic column results are reported based on seven 2019 columns (three at Sea ice and four at Onshore) and two 2018 columns (one at Sea ice and one at Onshore). Snow density was measured in 2018 at a vertical resolution of 3 cm using a snow cutter and a hanging scale. The snow density result is shown in Figure S2.

**2.3 Salinity measurements and ionic analysis**

All snow samples collected were transferred to 50 mL polypropylene tubes with screw caps (Corning CentriStar), which prior to field deployment had been rinsed with ultra-high-purity (UHP) water and dried in a class 100 clean laboratory in Cambridge, UK. All tubes with samples were put in a dark bag for temporary storage before moving into ice core boxes for storage and transportation. One set of snow samples were melted in the 0PAL laboratory to measure aqueous conductivity
using a conductivity meter (SensIon 5, Hach) with a measurement range of 0–200 mScm$^{-1}$ and a maximum resolution of 0.1 μScm$^{-1}$ at low conductivities (0–199.9 μScm$^{-1}$). Conductivity values were converted into psu, approximately equivalent to the weight of dissolved inorganic matter in grams per kilogram of seawater. Accuracy as stated by the manufacturer is ±0.001 psu at low salinities (<1 psu). Results are shown in Figure 2.

The 2018 snow samples were shipped frozen back to Cambridge, UK shortly after the campaign, and the 2019 samples
were shipped frozen directly to the Canadian Ice Core Lab (CICL) at the University of Alberta. All samples were only melted prior to the ion chromatography (IC) analysis, apart from a small portion of the samples that had been melted for salinity measurements. The 2018 samples were analysed in October 2018 and the 2019 samples were analysed in December 2019. Elevated salinity samples were diluted with UHP water, typically by a factor of 10 or 100 based on the estimated salinity. Due to the presence of fine particulates in the snow samples, all 2019 samples were filtered using Millex-GP Express PES
Membrane, Sterile, 33 mm, 0.22 μm filters (Merck Millipore Ltd., Cork, Ireland). The 2018 snow samples were analysed using



Thermo Scientific Dionex ICS-4000 ion chromatography systems, with ions of $Na^+$, $Ca^{2+}$, $Mg^{2+}$, $K^+$, $NH_4^+$, $Cl^-$, $Br^-$, $SO_4^=$, $NO_3^-$, $F^-$, acetate, formate, oxalate and MSA measured. The 2019 samples for IC analysis were run on a Dionex ICS-5000+ with ions of $Na^+$, $Ca^{2+}$, $Mg^{2+}$, $K^+$, $Cl^-$, $Br^-$, $SO_4^=$, $NO_3^-$, and MSA measured. Anion analysis was performed using an ionPac AS18-Fast-4µm column, and cation analysis was performed using an IonPac CS12A column.

Multiple samples (in 2019) were analysed to assess precision. The relative standard deviations of duplicate analyses, limits of detection (LOD, = 3 times standard deviation of filter blank average peak area), and limits of quantification (LOQ, = 10 times standard deviation of filter blank average peak area) for all sequences (~40 samples analysed per sequence) are reported in Table S1. The LOD of $Br^-$ is 0.200 µM with a relative standard deviation of 0.023 µM and the LOD of $NO_3^-$ is 0.484 µM with a relative standard of deviation of 0.037 µM. The mean statistical results for the ionic analysis of the 2018 and

2019 samples are given in Tables S2 and S3, respectively.  Mean values excluded outliers, defined as values more than 1.5 interquartile ranges above the upper quartile or below the lower quartile. Column means were calculated using values exclusively within the depth range ≥1.5 and ≤ 20 cm. Interpolation for vertical profile data consisted of 2-cm bin averages from 1.5-cm depth to the bottom of the snowpack.

## 2.4 MAX-DOAS measurements and BrO retrieval

Multi-axis Differential Optical Absorption Spectroscopy (MAX-DOAS) measurements of BrO partial columns were performed at the PEARL Ridge Lab. Spectra were recorded in the ultra-violet (UV) using a grating spectrometer (spectral resolution 0.45 nm) with a cooled (200 K) charge-coupled device (CCD) detector at 0.4–0.5 nm resolution. Elevation angles of 30°, 15°, 10°, 5°, 2°, 1°, and -1° were used in the elevation scans, and measurements were only taken with solar elevation above 4°. Differential slant column densities (dSCDs) of BrO and the oxygen dimer ($O_4$) were retrieved using the DOAS

technique with the settings described in Zhao et al. (2016) and Bognar et al. (2020). Reference spectra for the DOAS analysis were temporally interpolated from zenith measurements taken before and after each elevation scan. dSCDs were converted to profiles using a two-step optimal estimation method (Frieβ et al, 2011). First, aerosol extinction profiles were retrieved from $O_4$ dSCDs, and then the extinction profiles were used as a forward model parameter in the BrO vertical profile retrieval. The retrievals were performed for 0–4 km altitude on a grid with 0.2-km resolution. Due to the elevation of the measurement site,

the instrument often measures BrO in the free troposphere, except during strong wind episodes and storms that generate a deep boundary layer (Bognar et al., 2020).

## 2.5 Complementary datasets

There are two sets of local meteorology data used in this work: one from EWS (the archived data are available at Historical Data - Climate - Environment and Climate Change Canada (ECCC) (weather.gc.ca)) and one from the PEARL

Ridge Lab. In addition to the continuous datasets such as pressure, temperature and wind speeds, archived hourly report was used to derive daily weather conditions, such as blowing snow event, fog, ice crystal and trace precipitation. In addition,



ECMWF 6-hourly interim meteorological data (ERA-interim data) were used to explore large-scale weather conditions. Surface ozone measurements were made by a TEI 49i ozone analyzer deployed at 0PAL (Bognar et al., 2020). Hourly mean surface ozone data are available since the instrument was installed in late 2016. The UV index measured during the campaign period in 2018 and 2019 is shown in Figure S1, with data from the ECCC Brewer spectrophotometer (https://doi.org/10.1029/2004JD004820). In addition, NOAA back-trajectory output from the Hybrid Single-Particle Lagrangian Integrated Trajectory (HYSPLIT) model (Stein et al., 2015; Rolph et al. 2017) is used for diagnosing the air-mass history of selected events.

## 3 Results

### 3.1 Snow salinities

Figure 2 shows snow salinity distributions over sea ice (purple) and inland (orange) from all measurements, except for the tray samples. It can be seen that inland snow has a dual peak distribution with the first and second peaks appearing at 0.001–0.002 psu and 0.01–0.04 psu, respectively. On sea ice, snow has a triple peak distribution, with the first and second peaks overlapping with the inland peaks, indicating similar origins. The third peak at 0.2–0.4 psu clearly reflects sea water impacts.

Table 1 shows mean and median snow salinities (psu) in tray samples, at inland and sea ice sites, as well as in two snow types: soft fluffy snow and aged hard snow. Tray samples have the lowest mean value of 0.0070±0.0088 psu (N=14) which is lower than the inland mean (0.0290±0.113 psu, N=211) and the Sea ice mean (0.296±1.640 psu, N=146) by ~4 times and ~40 times, respectively. The lowest tray sample salinity of 0.00178 psu corresponded to a falling snow event on March 6, 2019 in a calm weather condition, and is close to the first peak salinity obtained in the surface layer snow, indicating the first peak of surface snow salinity (0.001-0.002 psu) is likely due to the precipitation dilution effect (due to less salt in falling snow). The tray samples median of 0.0035 psu is roughly one-third and one-tenth of the inland and sea ice samples median values (0.0115 and 0.0375 psu, respectively), but close to their second salinity peak, which is in line with the fact that the majority of tray samples are wind-blown particles.

The salinity difference between the two types of surface snow is significant. For example, at PEARL, the mean salinity of the soft fluffy snow is 0.0039±0.0029 psu (N=7), which is ~4 times smaller than that of the hard aged snow (0.0175±0.0046 psu (N=2)). At the Onshore site, the difference is ~11-fold (0.00327±0.00273 psu (N=73) vs. 0.0364±0.0112 psu (N=20)). At the sea ice site, the difference increases to ~23-fold (0.0105±0.0104 psu (N=44) vs. 0.2372±0.3836 psu (N=17)). Comparing these values with the snow salinity distributions in Figure 2, the soft fluffy snow salinity is seen to overlap well with the first peak, and the aged snow salinity overlaps well with the second peak. It indicates that fresh falling snow and the subsequent salt accumulation effect (due to water vapour loss by sublimation) are responsible for the first and the second salinity peak, respectively. The third salinity peak (0.2–0.4 psu) on sea ice is likely due to the sea water effect (due to upward migration of brine on sea ice), which is also observed in the Weddell Sea surface snow (Figure 16 in Frey et al., 2020). In





addition, the second snow salinity peak on sea ice (0.02–0.04 psu) is consistent with the Weddell Sea snow salinity on multi-
year sea ice, which indicates that the salts on multi-year ice surface layers could be a result of the accumulation effect for
deposited salts following the sublimation of water vapour rather than a direct sea water impact from the bottom (via the so-
called wicking migration effect). However, the Weddell Sea snow salinity does not resolve the first salinity peak at 0.001–
0.002 psu observed in Eureka, which could be due to the coarse vertical sampling resolution (2-3 cm) applied in their sampling.

Figure 3 shows surface snow salinity vertical profiles from the first layer (0–0.2 cm) to the third layer (0.5–1.5 cm),
and Figure S3 shows column salinity profiles. Note that tray samples salinity is shown in the upper panel of Figure 3. It can
be seen that salinity in the third layer is ~8 and ~15 times that in the first layer at the Onshore site and the Sea ice site,
respectively. The larger vertical gradient seen on sea ice is likely due to sea water influence from below. At PEARL, the
vertical trend is not clear, perhaps due to the very thin soft fluffy layer (only a few mm) and the thick crust layer observed at
the top of the hill where winds are stronger. Generally, tray samples salinity at the 0PAL site is on average larger than that at
the PEARL site; a similar result is also reflected in major ions, like $[Cl^-]$ and $[NO_3^-]$ (Figure 4 and S4). The relatively low
salinity at the PEARL site is likely attributed to the higher geographic altitude (~600 m) and the higher height of the mounted
tray above the ground (e.g., ~11 m at PEARL versus ~1 m at 0PAL).

The column salinity profiles in Figure S3 are predominantly 2018 data. Snow salinities at all inland sites do not vary
much with distance from the surface. PEARL has the lowest column mean salinity (0.0023±0.0019 psu). Onshore has >10
times the salinity (0.036±0.034 psu). The highest column mean snow salinity was observed on sea ice in 2018, with a mean
value (top 20 cm) of 1.673±2.09 psu, the maximum salinity of 18.73 psu was measured at the sea ice interface sample. It is
interesting to note that the 2019 column mean on sea ice (top 20 cm) is very low (0.085±0.026 psu), about 20 times lower than
the 2018 value, which is likely due to the dilution effect from the large iceberg grounded near Eureka.

The snow depth at the 2018 Sea ice sampling site is in the range of 24~28 cm, and a similar snow depth range (25-29
cm) was measured at the 2019 Sea ice site; this is partly because we deliberately chose a similar snow depth for sampling. In
addition, the measured precipitation amount between October 2017 and March 2018 is 20 mm, and the amount between
October 2018 and March 2019 is 19.4 mm, implying a similar snow depth on sea ice. Therefore, the significant difference in
column snow salinity between these two years cannot be due to snowpack depth difference, rather the difference could be due
to the saline supply at the sea ice interface. For example, the 2019 bottom snow (1-3 cm above the sea ice interface) salinity is
smaller than the 2018 bottom snow salinity by more than an order of magnitude (Figure S3), indicating a possible dilution
effect in 2019 from the iceberg grounded near EWS.

**3.2 Ion concentrations**

Figure 4 shows vertical profiles of 2019 snow ions $[Na^+]$, $[Cl^-]$, $[NO_3^-]$, $[Br^-]$, non-sea-salt bromide (noted as $nss[Br^-$
$]=[Br^-]_{obs}-0.0018\times[Na^+]_{obs}$), non-sea-salt $[SO_4^{2-}]$ ($nss[SO_4^{2-}]=[SO_4^{2-}]_{obs}-0.601\times[Na^+]_{obs}$) and enrichment factors of $Br^-$, $Cl^-$ and
$SO_4^{2-}$. Non-sea-salt values are calculated with the aim of removing salt effects on the concentration of bromine and sulphate,
which assists data interpretation particularly in comparisons between offshore and onshore sites as well as from different snow



depths. The enrichment factor is calculated following the equation of $EF_X=([X]/[Na]_{obs})/([X]/[Na]_{seawater})$, where $[X]/[Na]_{obs}$ represents the ratio of ion X to sodium in a sample, and $[X]/[Na]_{seawater}$ is the ratio in standard sea water (Wilson, 1975). If $EF_Y$ >1.0, ion X is enriched and if <1.0 it is depleted. To highlight the surface snow results, a lognormal Y-axis is applied. Tray

sample results are plotted in the top panel of each plot. Figure S4 shows the remaining profiles, including $[Ca^{2+}]$, $[Mg^{2+}]$, $[K^+]$, $[SO_4^{2-}]$ and enrichment of $[Ca^{2+}]$, $[Mg^+]$ and $[K^+]$.

As can be seen from Figure 4(a) and data in Table S3, the tray sample mean $[Na^+]$ (19.86±9.78 μM) at PEARL is 1.7 times of the first layer mean (11.80±5.20 μM), and at 0PAL, the tray sample mean $[Na^+]$ (36.99±23.25 μM) is 1.2 times of the first layer mean (31.33±34.37 μM). For $[Cl^-]$ (Figure 4(b)), the factor is 1.5 and 1.3 times at PEARL and 0PAL, respectively.

The enhancement of tray sample salts is likely due to accumulation effect following the water loss via sublimation processes. However, this accumulation effect cannot explain the even larger enhancement in $[NO_3^-]$ and nss$[Br^-]$ seen in Figure 4(c) and (e), respectively. For instance, at 0PAL, the tray sample mean $[NO_3^-]$ (3.41±2.05 μM) is 3.6 times the first layer mean (0.96±0.21 μM), at PEARL, the tray sample $[NO_3^-]$ (2.23±1.37 μM) is 1.8 times the first layer mean (1.24±0.50 μM). Eureka snow $[NO_3^-]$ is close to fresh snow nitrate of 2.5 μM at Alert in winter (Mcdonald et al., 2012), but smaller than snow nitrate

of ~7 μM at Barrow, Alaska (Krnavek et al., 2012).

For nss$[Br^-]$, at 0PAL, the tray sample mean (0.24±0.19 μM) is 2.4 times the first layer mean (0.10±0.07 μM). This indicates that airborne snow particles may uptake more gaseous nitric acid and soluble bromine species from the air than snow on the ground. As the deposition rate of chemical compounds to the ground is controlled by a series of transport steps: aerodynamic, sub-layer of the boundary and surface resistance (Wu et al., 1992).

Similar to snow salinity profiles (Figure 3), 2019 surface snow $[Na^+]$ (and $[Cl^-]$) increases significantly from the first layer to the third layer, e.g., by about 20-fold at Onshore, 30-fold at Sea ice, and 8-fold at PEARL (Figure 4(a) and (b)). The lowest sodium concentrations in the first layer are likely due to the precipitation dilution effect (due to less salt in falling snow particles). $[Br^-]$ (Figure4(d)) and $[SO_4^-]$ (Figure S4(d)) all show a similar vertical gradient, however nss$[Br^-]$ (Figure 4(e)) and nss$[SO_4^-]$ (Figure 4(f)) do not show such an increasing trend indicating the surface layer enhancement of the salts is largely

due to the accumulation effect. Moreover, the first layer nss$[Br^-]$ is generally higher than the second layer (0PAL is an exception), indicating the deposited bromide is from the air. A similar result is also in the bromine enhancement factor (Figure 4(g)). Regarding the 0PAL exception, this is mainly due to the two days samples collected were during and shortly after the precipitation event (on March 4 and 5, 2019).

The first layer $[Br^-]$ (Figure 4(d)) at Sea ice (0.40±0.20 μM (N=40)) and Onshore (0.40±0.17 μM (N=38)) are almost

the same, however, in the second and third layers, $[Br^-]$ at Sea ice (3.03±4.14 μM (N=51)) are significantly larger than that at Onshore (0.38±0.22 μM (N=58)) by more than an order of magnitude. When the sea water contribution is removed, the nss$[Br^-]$ concentration (Figure 4(e)) are not significantly different from each other (0.24±0.19 μM (N=32) vs 0.21±0.17 μM (N=50)), strongly indicating same atmospheric influence at the two sites.



The column mean nss[Br⁻] values at Sea ice is 0.22±0.18 μM (N=17), at Onshore it is 0.30±0.31 μM (N=89), which

are all positive, indicating a net sink of atmospheric bromine prior to the measurements. However, at PEARL, the positive

nss[Br⁻] was only observed in the tray samples (0.28±0.20 μM (N=21)) and the first snow layer (0.28±0.12 μM (N=31)). The

column mean nss[Br⁻] at PEARL is -0.05±0.08 μM (N=34) (Table S3), indicating snowpack at the top of the hill is bromide

depleted. Due to the lack of temporal variation information, the timing of the bromine depletion cannot be determined (e.g.,

before or after the precipitation) or more precisely whether it occurred soon after sunrise on February 21. The 2018 snow

samples at PEARL do not show a clear bromine depletion (Figure S5(d)), as the column mean nss[Br⁻] is slightly positive

(0.01±0.01 μM (N=8)) (Table S2). Snow bromide enrichments were reported at other Arctic sea level locations, e.g. in the

vicinity of Barrow, Alaska (Simpson et al., 2005), at Canadian Arctic Archipelago (Xu et al., 2016) and on first-year sea ice

(Peterson et al., 2019). However, at elevated sites in Svalbard (i.e. a few hundred meters above sea level), both bromide

enrichment (Spolaor et al., 2013) and depletion (Jacobi et al., 2019) were measured.

Figure 4(g-i) shows enrichment factors for Br⁻, Cl⁻ and SO₄²⁻ in 2019 snow samples. It can be seen that all these anions

are significantly enriched in surface layers and in tray samples, indicating important airborne sources. In particular, EF$_{Br^-}$ in

the tray samples, the first and second layers at the Onshore and Sea ice sites are larger than 10. Due to the lack of simultaneous

measurements of soluble inorganic bromine and filter aerosols, the dominant form of deposited bromide is unknown. Figure

S4 shows that cations [Ca²⁺], [Mg⁺] and [K⁺] are also enriched, especially in the bottom part at inland sites. In particular, [Ca²⁺]

enrichment factors at Onshore and PEARL sites are larger than 10, indicating strong terrestrial dust influence during the late

autumn when the land is not completely covered by snow.

Compared to 2019, the 2018 snow profiles (Figure S5) of sodium and bromide are much larger. For instance, 2018

column mean (1.5–20 cm) bromide on sea ice is 10.74±8.52 μM (N=80) (Table S2), in 2019, it is 6.47±5.36 μM (N=66) (Table

S3), but they are much smaller than mean 30.6 μM on thick first year ice (FYI) and 92.5μM on thin FYI at Barrow, Alaska

(Krnavek et al., 2012). However, the average [Br⁻] of 0.26 μM at Barrow inland is close to the Eureka inland values. The lower

2019 snow bromide on sea ice is likely due to the fresh water dilution by the grounded iceberg. However, surface snow bromide

does not follow this pattern; instead, the 2018 surface snow bromide is even lower than that of the 2019 values. For example,

bromide in the top 0.5 cm snow layer in 2018 is 0.23±0.10 μM (N=36), which is significantly lower than the 2019 value of

0.40±0.20 μM (N=40) in the 0–0.2 cm layer and the value of 3.03±4.14 μM (N=51) in the 0.2–0.5 cm layer. The lower 2018

surface snow bromide loading is likely related to the extremely low BrO partial columns measured in March at Eureka by

MAX-DOAS (Bognar et al., 2020), during which unusually calm weather, low aerosol optical depth (AOD) and coarse-mode

aerosol (likely SSA) concentrations were observed (see Section 3.3 and Figure 5 below for more details). These results indicate

that top layer snow bromide is largely controlled by atmospheric processes rather than by the underlying snowpack. This

conclusion is also consistent with previous finding that bromide concentrations at low salinities are dominated by atmospheric

exchange (Krnavek et al., 2012). Interestingly, surface layer nitrate concentrations between 2018 and 2019 are not significantly



different, e.g. the 2018 top 0.5 cm snow nitrate on sea ice is 3.13±1.00 μM (N=33), comparable to the 2019 first layer nitrate on sea ice of 3.46±1.55 μM (N=37).

### 3.3 Geographic heterogeneity of snow bromide and nitrate

Using the samples collected between February 26 and March 3, 2019, local geographic differences (across distance
of 1~2 m) of snow sodium, nitrate and bromide were assessed at each sampling site (Table S4). For bromide, the smallest heterogeneity is found at inland sites, particularly at PEARL, with the largest heterogeneity at Sea ice. For example, top 0.5 cm snow [Br⁻]=0.28±0.14 μM (nss[Br⁻]=-0.05±0.07 μM) at PEARL, compared to [Br⁻]=0.30±0.13 μM (nss[Br⁻]=0.25±0.13 μM) at Onshore and [Br⁻]=0.67±0.74 μM (nss[Br⁻]=0.43±0.48 μM) at Sea ice. Deeper layer snow bromide heterogeneity is generally larger than the upper layer (with an exception at PEARL), which is likely due to the large uncertainty of accumulated
bromide. The smallest standard deviation of nss[Br⁻] is at PEARL (0.075 μM), with the medium 0.21 μM at Onshore, and the largest 0.73 μM at Sea ice. Nitrate in the top 0.5 cm and the 0.5-1.5 cm layer are not significantly different, indicating they are independent of snow salts. The top 1.5 cm mean [NO₃⁻] at Sea ice is 3.62±1.34 μM, at Onshore is 2.95±0.86 μM, and at PEARL is 2.03±0.43 μM. Similar to bromide, PEARL has the smallest mean value and uncertainty. Note that the source of uncertainty is not solely from geographic variation; other factors such as temporal variations (see Section 3.4) as well as the
bias in depth estimation all contribute to the uncertainty.

On March 4 and 5, 2019, snow samples were collected during a precipitation event (Figure 6a) from three sub-layers (0-0.2, 0.2-0.5, and 0.5-1.5 cm) at the 0PAL, Onshore, and Sea ice sites (also in Table S4). The 0.2 mm precipitation measured meant a ~1 cm snowfall on the surface, which explains the low concentrations and low variability of [Br⁻] at Onshore. Moreover, the top 0.2 cm snow [Br⁻] (0.12±0.00 μM) at Onshore is very close to that for 0PAL (~5 km away) (0.14±0.02 μM),
indicating they are under the same atmospheric influence. However, at Sea ice, the first layer [Br⁻] (0.38±0.04 μM) is ~3 times that of the onshore value, highlighting the underlying sea ice effect. The sea ice effect is more significant in the second (0.2-0.5 cm) layer, where high [Br⁻] (5.73±5.57 μM) was measured. However, the corresponding nss[Br⁻] (0.01±0.04 μM) in the second layer at Sea ice is very low, and close to the nss[Br⁻] (0.01±0.00 μM) at 0PAL and at Onshore (0.09±0.03 μM), also indicating the same atmospheric influence. For nitrate, the precipitation effect is less significant; the sea level mean [NO₃⁻]
(3.28±1.10 μM) is very close to the Sea ice mean obtained during February 26-March 3. The sea level nitrate is also higher than the hilltop mean of 2.03±0.43 μM, indicating a vertical gradient of atmospheric nitrogen oxide between the boundary layer and the free troposphere.

### 3.4 Time series of surface snow [Br⁻] and [NO₃⁻]

Figure 5 shows the 2018 time series of local meteorology (a-b), surface ozone at 0PAL and 0-4 km BrO partial column
(c), and top 0.5 cm snow [Na⁺] (d), [NO₃⁻] (e), [Br⁻] (f), and nss[Br⁻] (g) at the Sea ice, Onshore, and PEARL sites. Figure 6 shows the 2019 time series of meteorology (a-b), surface ozone at 0PAL and 0-4 km BrO partial column (c), and tray samples



[Na$^+$] (d), [NO$_3^-$] (e), [Br$^-$] (f), and nss[Br$^-$] (g) at the 0PAL and PEARL sites. Figure 7 shows the 2019 time series of surface snow nitrate (a-c) and non-sea-salt bromide (d-f) in three sub-layers: 0–0.2 cm, 0.2–0.5 cm, and 0.5–1.5 cm.

Extremely calm conditions were observed in March 2018, with wind speeds <5 m s$^{-1}$ most days. Figure 5(a) shows strong inversions between EWS and PEARL in March, e.g., the temperature difference between these two heights can be >10°C. Blowing snow events were only recorded on March 3 and 5, 2018 which is unusually infrequent. On the contrary, March 2019 was very windy, with blowing snow events recorded on March 1, 2, 4, 12–14, 18, 19, 23–25, and 28, 2019, approximately 40% of the days.

March 2018 had a very low background BrO partial column of ~1×10$^{13}$ molecules cm$^{-2}$ or less (Figure 5(c)), while March 2019 had a background BrO partial column almost two times the 2018 level (Figure 6(c)). Accordingly, surface ozone concentrations in March 2018 were generally higher than that in March 2019. For example, the background surface ozone in March 2018 was mainly around 30 ppbv, in March 2019, the background surface ozone is mainly below 20 ppbv indicating accelerated ozone losses due to enhanced BrO loading in the air.

Here we focus on the 2019 datasets (Figures 6 and 7) for further discussion. The meteorological record indicates that fog events were recorded on March 7, 15, 17-20, 22, 23 and 28, 2019. Some of these events were accompanied by precipitation (daily amount ≥0.2 mm, as shown in Figure 6(b)). Precipitation events were recorded on March 5, 6, 7, 10, 15, 19, 27, 28, 30, and 31 with a total monthly precipitation of 2 mm. On average, precipitation occurs at a frequency of every ~3 days, which is consistent with the average Arctic snow age used in Huang and Jeaglé (2012). In addition, trace precipitation events are included, occurring on March 1, 2, 4, 5, 6, 10-13, 15, 18-21, 24, and 28 (~50% of the time), then the average precipitation frequency is reduced to every 1.5 days.

Tray sample sodium has a large day-to-day variability (Figure 6(d)). The low sodium concentrations measured on March 6 and 11, 2019 are likely due to the precipitation dilution effects, and the high sodium concentrations measured on March 4–5, 13–14, and 24 are likely related to the windy conditions. In general, 0PAL tray sample sodium does not show a clear increase trend with time, though this is evident at PEARL.

Tray sample nitrate at 0PAL shows a clear increasing trend (Figure 6(e)) with a mean slope of 0.177±0.073 μM d$^{-1}$ (R=0.46, *p*=0.020, N=24) (Table S5). At Sea ice, snow nitrate in the first layer (0–0.2 cm) has a slope of 0.253±0.101 μM d$^{-1}$ (R=0.50, *p*=0.022, N=21), and at Onshore, it is 0.285±0.124 μM d$^{-1}$ (R=0.48, *p*=0.033, N=20). In the second layer (0.2–0.5 cm), snow nitrate slope at Sea ice is 0.235±0.054 μM d$^{-1}$ (R=0.70, *p*=0.0003, N=22) and at Onshore it is 0.165±0.063 μM d$^{-1}$ (R=0.52, *p*=0.017, N=21). In the third layer (0.5–1.5 cm), snow nitrate slopes at Sea ice and Onshore are smaller, 0.057±0.025 μM d$^{-1}$ (R=0.41, *p*=0.027, N=29) and 0.08±0.027 μM d$^{-1}$ (R=0.51, *p*=0.007, N=27), respectively. These slope values are only 1/5 to 1/3 of the top two-layer values, indicating a reduced nitrate deposition flux to deeper snow layers. The standard deviations of nitrate slope at sea level are ½ to ¼ of the mean slope values indicating the linear regression fits are statistic significant.



Nitrate at PEARL behaves differently. For instance, a near zero increasing trend was observed in PEARL tray samples
and in the first layer. Moreover, a negative slope was obtained in the second and third layers, respectively. These results
indicate that deposition flux at the top of the hill is reduced and cannot compensate for the nitrate loss via photolysis. The
positive slope at the sea level indicates the deposited nitrate during the ~1 day period was larger than the photochemical loss
during daytime.

Surface snow [Br⁻] and nss[Br⁻] show a very similar increasing trend (Figure 6(f) verse 6(g)), this is due to the large
bromine enrichment factor or weak sea water impact. The 2019 tray sample nss[Br⁻] slope at 0PAL is 0.023±0.006 µM d⁻¹
(R=0.64, $p$<0.001, N=24 ) and at PEARL it is 0.013±0.006 µM d⁻¹ (R=0.56, $p$<0.04, N=14) (Table S5). Figure 7(d) shows the
first layer nss[Br⁻] slope at Sea ice is 0.024±0.0096 µM d⁻¹ (R=0.50, $p$=0.020, N=21), and at Onshore it is 0.023±0.008 µM d⁻¹ (R=0.56, $p$=0.011), and at PEARL it is 0.012±0.004 µM d⁻¹ (R=0.65, $p$=0.005, N=20). The second layer nss[Br⁻] slope (Figure
7(e)) is slightly smaller: 0.017±0.012 µM  d⁻¹ (R=0.42, $p$=0.18, N=12) at Sea ice, 0.038±0.009 µM d⁻¹ (R=0.38, $p$=0.13, N=17)
at Onshore, and 0.017±0.007 µM d⁻¹ (R=0.66, $p$=0.036, N=10) at PEARL. Due to the few measurements in the third layer,  a
robust trend could not be derived, while Onshore dataset indicates a near zero slope (0.003±0.007 µM d⁻¹ (R=0.11, $p$=0.63,
N=23, Figure 7(f)). In the first and second layers, standard deviation values are about ½ to ¼ of the slope values, indicating
the bromide trends derived are statistic significant. Due to the large uncertainty, no clear trend (i.e. a zero slope) was obtained.

In addition to the long-term trend, both nitrate and bromide show a large day-to-day perturbation. For instance, the
maximum nitrate concentration of >15 µM was observed on March 18, 2019 in both tray samples and the first layer snow,
which is likely associated with a heavy fog event, lasing more than 16 hours with visibility dropped from >10 km before the
fog event to only 1~2 km. Meanwhile, snow bromide also showed an enhancement, e.g. with concentrations >1 µM measured
at the Sea ice and Onshore sites (Figure 7(d)). Another large bromide enhancement event was observed in tray samples on
March 22, 2019, it is also associated with a >6 hours fog event. March 15, 2019 experienced the longest fog event (>17 hours),
however, bromide and nitrate did not show any enhancements, which could be related to the precipitation effect, as a 0.2 mm
precipitation was recorded.

In general, there is not a clear correlation between surface snow sodium and bromide at Eureka. However, on March
4, 14 and 24, 2019 when it was very windy, high bromide and sodium concentrations were observed, indicating a blowing
snow sourced sea salt contribution.

As noted above, March 18, 2019 was a heavy fog day. The signals of enhanced snow nitrate can be detected in tray
samples and the first layer, and is still slightly detectable in the second layer at the Onshore site (Figure 7(a-b)). However, the
enhancement signal disappears in the third layer, indicating the fog-related nitrate deposition is mainly confined to the top 0.5
cm snow layer.

The 2018 time series dataset shows a similar story. For example, top 0.5 cm snow nitrate at the Sea ice site has a slope
of 0.240±0.032 µM d⁻¹ (R=0.93, p<0.001, N=11, Table S5) and at the Onshore site it is 0.166±0.073 µM d⁻¹ (R=0.61, $p$=0.047,
N=11). However, 2018 snow [Br⁻] and nss[Br⁻] do not show a clear increasing trend (Figure 5(f, g) and Table S5). The slope





at Onshore is very small and not significant (0.005±0.005 μM d⁻¹, R=0.27, N=11), indicating a weak bromide deposition flux. Although the 2018 snowpack column bromide on sea ice is several times the 2019 column mean (Tables S2 and S3), the small bromide deposition flux in 2018 is likely due to the calm weather and the extremely low BrO loading as measured by MAX-

DOAS (Bognar et al., 2020).

**3.5 Morning versus afternoon nitrate and nss[Br⁻]**

Compared to morning samples, afternoon samples at Eureka undergo 3-7 more hours of sunlight, which means photochemical loss of nitrate and bromide from snowpack may be enhanced as a result. The 2019 morning and afternoon concentrations of nitrate and nss[Br⁻] are shown in Figure 8. The mean [NO₃⁻] for morning samples (at Sea ice and Onshore)

is 3.02±1.56 μM, which is larger than the afternoon mean of 2.79±1.45 μM by 0.23 μM; at the PEARL site, the photochemical loss is 0.48 μM (1.66±0.48 μM in the morning vs. 1.18±0.47 μM in the afternoon), with the largest change of 1.48 μM (2.11±0.22 μM in the morning vs. 0.63±0.10 μM in the afternoon) occurring in the first layer.

Snow bromide also shows a similar photochemistry effect, however the signals are not significant across all sampling sites. For example, at the Onshore site, the morning, nss[Br⁻] in the first layer is 0.25±0.12 μM, which is larger than the

afternoon 0.23±0.21 μM by 0.02 μM; in the second and third layers, the morning-afternoon differences are not significant due to large standard deviations. At the Sea ice site, the morning-afternoon differences are all within the uncertainties, e.g. the morning nss[Br⁻] in the first layer is even smaller than the afternoon (by 0.01 μM); in the second layer, the morning nss[Br⁻] (0.18±0.03 μM) is larger than the afternoon (0.16±0.07 μM) by 0.02 μM; in the third layer (0.5-1.5 cm), the morning nss[Br⁻] (0.18±0.03 μM) is larger than the afternoon value (0.14±0.04 μM) by 0.04 μM. Based on above numbers, a mean daytime

bromide loss rate of 0.027 μM at sea level was obtained. At PEARL, the photochemistry effect is only obtained in the first layer, with a morning-afternoon change of 0.07 μM (0.22±0.13 μM vs 0.15±0.03 μM).

However, the tray samples of [NO₃⁻] and nss[Br⁻] responded differently, with morning concentrations generally lower than their afternoon values. For example, the morning mean [NO₃⁻] for the tray samples (2.09±1.32 μM) is smaller than the afternoon mean (3.56±2.54 μM) by 1.47 μM. For bromide, the morning mean nss[Br⁻] for the tray samples (0.21±0.16 μM) is

also smaller than the afternoon mean (0.36±0.31 μM) by 0.15 μM. The afternoon enhancement means there is a strong net uptake of nitrate and bromide by airborne particles, resulting in a deposition rate that is larger than the photochemical loss rate, in contrast to the relationship for the surface snow particles. This finding is consistent with the vertical profiles of nitrate and bromide shown in Figure 4(c) and (e), where the tray sample [NO₃⁻] at 0PAL is 3.6 times the first layer nitrate, and at PEARL it is 2.1 times the first layer nitrate. Additionally, the tray sample nss[Br⁻] at 0PAL is 2.6 times the first layer value.

**3.6 Deposition flux of bromide and nitrate**

The daily slopes of nitrate and bromide concentrations derived above can be used to calculate their deposition flux to snowpack following this new equation:



$$Flux = \frac{A}{T}\sum_{k=1}^{n} S_k H_k D_k \qquad\qquad (R3)$$

where *Flux* is mean net deposition flux (in units of molecules cm$^{-2}$ s$^{-1}$) over the observational period from snow layer 1 to n, *A* is Avogadro's number of gas ($6.02\times10^{23}$ molecules mole$^{-1}$), *T* is seconds in 1 day (86400 s d$^{-1}$), $S_k$ is the derived daily slope in snow layer *k* (in μM d$^{-1}$), $H_k$ is the corresponding snow layer depth (in cm), and $D_k$ is snow density of the layer (in g cm$^{-3}$).

In this study, n=3. A low snow density of 0.15 g cm$^{-3}$ is used for the top two layers, and 0.3 g cm$^{-3}$ is used for the third layer. For nitrate, a mean slope value at sea level (from the Sea ice and Onshore sites) of 0.27, 0.2, and 0.07 μM d$^{-1}$ were used in the first, second, and third layers, respectively. Therefore, an integrated nitrate deposition flux of $2.6\times10^8$ molecules cm$^{-2}$ s$^{-1}$ from the top 1.5 cm snow is obtained. At PEARL, the integrated deposition flux is negative ($-1.0\times10^8$ molecules cm$^{-2}$ s$^{-1}$) according to the mean slope of 0.0, -0.013, and -0.04 μM d$^{-1}$ in the three sub-layers. These results indicate that surface snow at sea level is a net sink of atmospheric nitrate, and at the top of the hill is a source of reactive nitrogen. Our derived nitrate deposition flux at sea level Eureka is close to the winter average flux of $2.7\times10^8$ molecules cm$^{-2}$ s$^{-1}$ derived at Alert, Nunavut (Macdonald et al., 2017) and $\sim4\times10^8$ molecules cm$^{-2}$ s$^{-1}$ at Svalbard (Björkman et al., 2013), justifying the method used in this work.

For bromide, the integrated deposition flux is $1.01\times10^7$ molecule cm$^{-2}$ s$^{-1}$ at sea level, using a mean slope of 0.024, 0.016, and 0.0 μM d$^{-1}$ in the three sub-layers, respectively. At PEARL, the integrated flux is $7.9\times10^6$ molecules cm$^{-2}$ s$^{-1}$ which is $\sim$20% lower than at sea level. This small vertical gradient strongly indicates that BrO concentrations (and total inorganic bromine species) at sea level and in the free troposphere are not significantly different at Eureka, which is in agreement with the conclusion in Bognar et al. (2020). This implies that either bromine at Eureka is mixed well in the lower troposphere (mainly during strong winds with enhanced BrO) or local snowpack at sea level is not a large source of reactive bromine. As mentioned previously, from winter to early spring, Eureka boundary layer is very shallow and stratified in calm conditions, thus most of the time PEARL is located in the free troposphere. Therefore, if local snowpack on sea ice in the fiord is a large source of reactive bromine, an enhanced deposition flux at sea level should be detected. In addition, previous work focusing on atmospheric chemistry has demonstrated that large BrO enhancement events observed in Eureka in early springtime are mostly transported via cyclones (Zhao et al., 2016; 2017; Yang et al., 2020). The transported bromine in association with storms means well-mixed bromine species from the surface up to the free troposphere (>1 km), which explains the small vertical gradient of deposited bromine flux in this current work.

## 3.5 Relationship between surface snow [NO$_3^-$] and [Br$^-$]

There are multiple sources of snowpack bromide and nitrate. For example, bromide may come from reactive bromine gases (such as HOBr, BrO and BrONO$_2$) and the terminal product HBr in both gas phase and particle phase. Due to the lack of in-situ data, we could not accurately quantify the contribution of HBr to snow bromide at Eureka. However, a modelling work (focusing on Antarctic coastal Dumont d'Urville chemistry) indicates that reactive bromine species dominate total gaseous inorganic bromine. For instance, gas phase HOBr and BrO together account for $\sim$2/3 of total inorganic bromine on





average, and gas phase HBr only accounts for 12%, In austral spring (September-October) HBr partitioning is higher, but does not exceed 25% (from Figure 14 of Legrand et al., 2016). Bromine may accumulate as gas phase HBr when ozone depletion has terminated (Lehrer et al., 2004), but during the campaign period, surface ozone rarely dropped below 2-3 ppbv (Figures 5(a) & 6(a)), therefore, gas phase HBr accounts for a small fraction in total inorganic bromine. In addition, airborne particles can take up gas phase HBr from the air. Impactor data from both hemispheres indicate that the smallest particles (sub-micron

size mode) are normally enhanced in Br⁻ (as compared to sea salt reference), while large-sized particles are slightly Br- depleted (Alvarez-Aviles et al., 2007; Legrand et al., 2016). If HBr in sea-salt particles dominates Br⁻ in surface snow, then a relationship between sodium and bromide should exist, however, this relationship is not detected in our dataset (not shown); this is also in line with the finding at coastal Alaska (Simpson et al., 2005). Moreover, the observed large bromide enhancement factor (>10, Figure 4g and Section 3.2) strongly indicates that bromide in surface snow is not related to sea salt. Thus, it is reasonable to

make an assumption that nss-Br⁻ mainly comes from reactive bromine species.

Relatively low nitrate concentrations of 0.1-8.2 µM were detected in Arctic sea ice (Clark et al., 2020); their isotope-based investigation of the origin of nitrate indicates that the atmospheric contribution accounts for 40% or less of the sea ice nitrate, indicating the important of atmospheric reactive nitrogen to sea ice nitrate. Our data indicate that there is no significant relationship between sodium and nitrate (not shown) in surface snow, which is consistent with the finding for Alaska (Krnavek et al., 2012).

However, a significant relationship is found between surface snow [NO₃⁻] and [Br⁻] (Figure 9) in tray samples at 0PAL, 0–0.2 cm, and 0.2–0.5 cm layer snow at the Onshore site (2019), and the top 0.5 cm snow at the Onshore site (2018), with coefficient R in the range of 0.4-0.7. This relationship remains when nss[Br⁻] is used in the analysis with a similar R of 0.23–0.66 (Figure S6). Moreover, the ratio of [NO₃⁻]/[Br⁻] ranges from 3.5–6.8, indicating that one molecule bromide deposited to surface is

likely accompanied by 4–7 nitrate molecules. For the first time, we see field evidence on a time scale of one day showing this effect such as via reactions of (R1) and (R2). This finding further confirms previous conclusions regarding the role that reactive bromine plays in determining high latitude atmospheric reactive nitrogen (e.g., Yang et al., 2005; Morin et al., 2008). Such a relationship is not seen in surface snow on sea ice, likely due to sea water effect on bromide. However, we do see a weak correlation between them at PEARL (not shown).

**4 Discussion and implications for polar chemistry**

Surface snow nitrate and bromide in the morning samples are higher than in the afternoon samples indicating a strong snow photochemistry effect, which is in agreement with numerous previous studies regarding snowpack as a direct source of reactive nitrogen and bromine under sunlight. However, the measured mean net nitrate loss rate during the daytime of 0.23 µM at the sea level site and 0.48 µM at the hilltop site is almost an order of magnitude larger than the bromide loss

rate of 0.027 µM at sea level and 0.07 µM at the hilltop. This result strongly indicates that the reactive bromine emission flux from snowpack might be smaller than the reactive nitrogen emission flux by an order of magnitude.



Snow nitrate can be directly photo-dissociated under sunlight, while snow bromide activation needs heterogeneous photochemistry. Therefore, photons are a necessary but not a sufficient condition for bromine recycling. For example, the heterogeneous reactions for snowpack bromide reactivation involve three transport steps: aerodynamics brings HOBr gas to

the near surface sub-layer, and the following transport requires HOBr molecules to pass through the quasi-laminar boundary layer before they can be eventually absorbed by snow particles. It has been shown that the above three processes vary greatly, depending on the depositing species and surface characteristics (Wu et al., 1992). The relatively slow bromide loss rate observed in surface snow implies the bromide heterogenous reactions are somehow rate-limited, likely by the deposition flux of hypohalous acid such as HOBr or other reactive bromine species such as BrONO$_2$.

The mean daytime loss of nitrate and bromine at sea level is 0.23 μM and 0.027 μM, respectively, which is close to the daily deposited nitrate and bromide amounts of 0.27 μM and 0.024 μM, respectively. If we assume a constant deposition flux for both nitrate and bromide to surface snow, and a mean 5-hour interval between the afternoon and morning sampling, then we can derive an absolute deposition flux of 0.632 μM d$^{-1}$ for nitrate and 0.064 μM d$^{-1}$ for nss[Br$^-$], which is 2.3 and 2.7 times their net deposition flux, respectively. Meanwhile, the absolute photochemistry loss of nitrate is 0.362 μM and of bromide

is 0.040 μM, which is 1.6 and 1.5 times the net loss rate, respectively. Applying the corrected flux (a factor of 2.3) to the snow nitrate slopes results in a larger deposition flux of 5.98×10$^8$ molecules cm$^{-2}$ s$^{-1}$; similarly, applying a correction factor of 2.7 to the bromide slopes results in an updated nss[Br$^-$] deposition flux of 2.73×10$^7$ molecule cm$^{-2}$ s$^{-1}$.

Snowpack is thought to be a highly permeable material, meaning gasses and fine aerosols could penetrate into deep layers (Harder et al., 1996; Björkman, et al., 2013) due to the exchange of air with the atmosphere (Sturm and Johnson, 1991;

Albert and Hardy, 1995; Colbeck, 1997; Albert et al., 2002), however, our data show that most deposited species were in the top 0.5 cm layer. For example, at sea level nitrate slopes reduce significantly from the first layer mean of 0.26 μM d$^{-1}$ to the second layer mean of 0.20 μM d$^{-1}$ and third layer mean of 0.07 μM d$^{-1}$; a similar trend is also obtained for nss[Br$^-$] slopes, with the first layer slope of 0.023, the second layer slope of 0.014 and the third layer slope of 0.0 μM d$^{-1}$. These results indicate deposited nitrate and bromide are largely confined to the skin layer. The fog-related bromide and nitrate enhancements are

only found in the top two layers (<0.5 cm), which is in agreement with conclusions of Domine et al. (2004) who state that the aerosol effect on snow ion concentrations is limited to the top few cm. In extreme conditions, applying the above derived absolute nitrate deposition flux of 0.632 μM d$^{-1}$ and the absolute nss[Br$^-$] deposition flux of 0.064 μM d$^{-1}$ to all three layers, then a nitrate deposition flux of 1.65×10$^9$ molecules cm$^{-2}$ s$^{-1}$ and a bromide deposition flux of 1.72×10$^8$ molecules cm$^{-2}$ s$^{-1}$ can be calculated. These may represent the upper limits of the deposition flux of nitrate and bromine.

If the deposited nitrate and bromide to the surface snow are assumed to be roughly balanced by local snowpack emissions, then the above derived fluxes can be used to estimate snowpack emission fluxes. In this case, the local snowpack on sea ice may have a reactive nitrogen emission flux of (2.6-5.98)×10$^8$ molecules cm$^{-2}$ s$^{-1}$ with an upper limit of 1.65×10$^9$ molecules cm$^{-2}$ s$^{-1}$, which are well in the range of previously measured NOx emission fluxes (Jones et al., 2001; Zhou et al., 2001; Honrath et al., 2002; Beine et al., 2002; 2003; Oncley et al., 2004; Frey et al., 2013; Chan et l., 2018). However, the



local snowpack reactive bromine emission flux of $(1.01\text{-}2.73)\times10^7$ molecules $cm^{-2}$ $s^{-1}$ is smaller than the measured emission fluxes of $(0.7\text{–}12)\times10^8$ molecules $cm^{-2}$ $s^{-1}$ (Custard et al., 2017) by 1-2 orders of magnitude. The upper end emission flux of $1.72\times10^8$ molecules $cm^{-2}$ $s^{-1}$ is close to the lower end of the measured flux. If this is the case, then local snowpack emission should not cause the boundary layer BEEs and ODEs observed at Eureka, rather it may affect the background BrO.

As shown above, the uncertainty of the bromide measurements is larger in the third layer and on sea ice, where the
accumulated salts and sea water make the non-sea-salt bromine signals harder to detect. We also do not have samples from deeper snow layers (>1.5 cm), therefore our measurements may underestimate the deposition flux of nitrate and bromide. Although the snow e-folding depth (light attenuation) at Eureka was not measured, previous measurements at other polar sites indicate that it varies from a shallow 2-5 cm (Erbland et al., 2013) to a deep 10-20 cm (France et al., 2011) in Antarctica. At Cambridge Bay, Canada, an e-depth of 16 cm was reported in March snowpack (Xu et al., 2016). Therefore, the loss of nitrate
and bromide via photochemistry or the release of reactive nitrogen and bromine may come from a deeper snow layer, which we have no data to confirm.

In addition, the method used to derive the deposition flux of nitrate and bromide is different from the instrument-based measurements of gas reactive nitrogen and bromine. For bromine, the method is largely dependent on the choice of sampling location, ideally where the snowpack on sea ice and at inland should be less disturbed by other bromine sources such
as open leads, polynyas and sea spray. Therefore, this method may not work well in the area where sea ice has a significant amount of mobility, with sea ice opening and closing frequently. The conclusion derived in this study may only representative of local characteristics, as sea ice conditions at Eureka are quite different from those in the central Arctic (Shupe et al., 2022). However, the physical and chemical processes involved in bromide deposition and reactive bromine release may remain the same across locations. To confirm this, a more comprehensive field campaign under typical Arctic sea ice conditions is needed.
Note that the lifetime of an individual reactive bromine species such as BrO and HOBr is short (only a few minutes) under sunlight; however, the quick recycling via photochemically heterogeneous reactions to convert inactive HBr back to the active form, such as $Br_2$ or BrCl, means that as a family, the lifetime of total inorganic bromine species is in fact much longer (e.g., from a tropospheric mean of 4-5 days (Yang et al., 2005) to >10 days (von Glasow et al., 2004)). For nitrogen oxides ($NO_X$), the lifetime in the Arctic springtime is 2-6 days (Stroud et al., 2007). It is important to note that within a stable boundary
layer, the typical time needed for a surface signal to reach the upper layer is 7-30 hrs (Stull, 1988). Therefore, within the one-day timescale selected for snow sampling, emitted reactive bromine and nitrogen should have sufficient time to mix well in the boundary layer and reach a quasi-equilibrium state with other processes, including deposition and photochemistry.

## 5 Conclusions

Based on two years of daily surface snow sampling in the Canadian high Arctic, an integrated spring nitrate deposition
flux of $2.6\text{-}5.98\times10^8$ molecules $cm^{-2}$ $s^{-1}$ has been derived from the top 1.5 cm snow in the fiord of Eureka. At the top of the hill (PEARL Ridge Lab, ~600 m), nitrate deposition flux is negative ($-1.0\times10^8$ molecules $cm^{-2}$ $s^{-1}$) indicating snow is losing nitrate



in early spring. Integrated bromide deposition flux at sea level is $1.01\text{-}2.73\times10^7$ molecules $cm^{-2}$ $s^{-1}$; at the hilltop, the deposition flux is ~20% smaller. The small vertical gradient between the boundary layer and the free troposphere indicates local snowpack is a weak reactive bromine emission source. On the contrary, the large vertical gradient in nitrate deposition flux strongly
indicates that local snowpack is a large emission source of reactive nitrogen. In addition, the bromide deposition flux at sea level is more than an order of magnitude smaller than the nitrate deposition flux.

Surface snow nitrate and bromide in the morning samples are generally higher than in the afternoon samples, highlighting a significant snow photochemistry effect. The mean daily photochemistry loss is 0.23-0.362 μM for nitrate, and 0.027-0.040 μM for bromide, which implies that the reactive bromine emission flux from the snowpack should be smaller than
the reactive nitrogen emission flux by an order of magnitude. This emission flux difference is consistent with the one order of magnitude difference in deposition flux derived in this study, justifying the assumption that emitted reactive bromine and nitrogen should be roughly balanced by deposited bromide and nitrate, respectively. Therefore, we conclude that the local snowpack at Eureka is a weak source of reactive bromine and thus unlikely to be the source of BEEs or ODEs observed locally. However, due to the lack of field data from other Arctic locations, we cannot conclude robustly whether the result obtained in
this study is a local characteristic or can be extended to a broad Arctic area. However, our finding is in line with the conclusion made by Legrand et al. (2016) that snowpack bromine emission is not important over the Antarctic Plateau.

Additionally, the surface snow (<0.5 cm) nitrate and bromide are found to be significantly correlated with a $[NO_3^-]/[Br^-]$ ratio of 4–7. This means that reactive bromine could effectively accelerate $NO_X$-to-nitrate conversion. This is the first time such an effect has been seen on a timescale of one day. This also reinforces the importance of reactive bromine in polar
and high latitude reactive nitrogen budgets, and its atmospheric oxidising capacity.

**Author Contributions**

XY designed the field experiment and performed snow sampling, salinity measurements, and data interpretation. KS and KAW co-organised the campaign. PF and the Canadian Network for the Detection of Atmospheric Change (CANDAC) team provided logistics support and performed snow sampling. AC led ion chromatography analysis for the 2019 samples. KB
provided MAX-DOAS BrO data, SMM and PE provided surface ozone data, and XZ supplied local meteorology and radiation data. MSG performed major ionic analysis and plotting. XY led the writing with contributions from all co-authors.

**Competing Interest**

The authors declare that they have no conflict of interest.



**Data Availability**

All the data will be archived in BAS Polar Data Centre.

**Acknowledgements**

We thank the UK NERC Arctic office for their support of this study via two UK-Canada bursary programs: "The role of tundra snowpack chemistry in the boundary layer bromine budget at Eureka, Canada" (2018), and "A second investigation of the role of tundra snowpack chemistry in the boundary layer 'bromine explosion'" (2019). The Eureka MAX-DOAS BrO
measurements were made at the PEARL Ridge Laboratory by CANDAC, primarily supported by NSERC, CSA, and ECCC. The UV index data are from Brewer spectrophotometer run by Environment and Climate Change Canada (ECCC). We thank CANDAC and ECCC for enabling and supporting the snow sampling campaigns and the BAS Ice Core Laboratory for analysing the 2018 samples (by Sara L. Jackson). The NOAA Arctic Research Program, Physical Sciences Laboratory, and Global Monitoring Laboratory have contributed to establishing surface ozone measurement programs in Eureka.

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



Table 1. Mean and median snow salinities (psu) in various snow samples: in tray, at inland and sea ice sites. Surface snow (<0.5 cm) salinities are in two snow types: fluffy soft snow and aged hard snow.

| | Snow types | Sample number | Year | Mean ± 1 standard deviation | Median |
|---|---|---|---|---|---|
| Tray samples | all | 14 | 2019 | 0.0070±0.0088 | 0.0035 |
| Inland samples[a] | all | 211 | 2018, 2019 | 0.0290±0.1130 | 0.0115 |
| Sea ice samples[b] | all | 146 | 2018, 2019 | 0.2960±1.6400 | 0.0374 |
| PEARL surface | fluffy soft | 7 | 2018 | 0.0039±0.0029 | 0.0038 |
| | aged hard | 2 | 2018 | 0.0175±0.0046 | 0.0175 |
| Onshore surface | fluffy soft | 73 | 2018 | 0.0033±0.0027 | 0.0021 |
| | aged hard | 20 | 2018 | 0.0364±0.0112 | 0.0375 |
| Sea ice surface | fluffy soft | 44 | 2018 | 0.0105±0.0104 | 0.0057 |
| | aged hard | 17 | 2018 | 0.2372±0.3836 | 0.0896 |

[a] Inland data contain all salinity measurements for snow samples in the surface layers and columns collected at the Onshore, 0PAL/Creek, PEARL and airport sites. [b] Sea ice data contain all salinity measurements for samples in the surface layers and columns collected over sea ice (see Section 2.2).




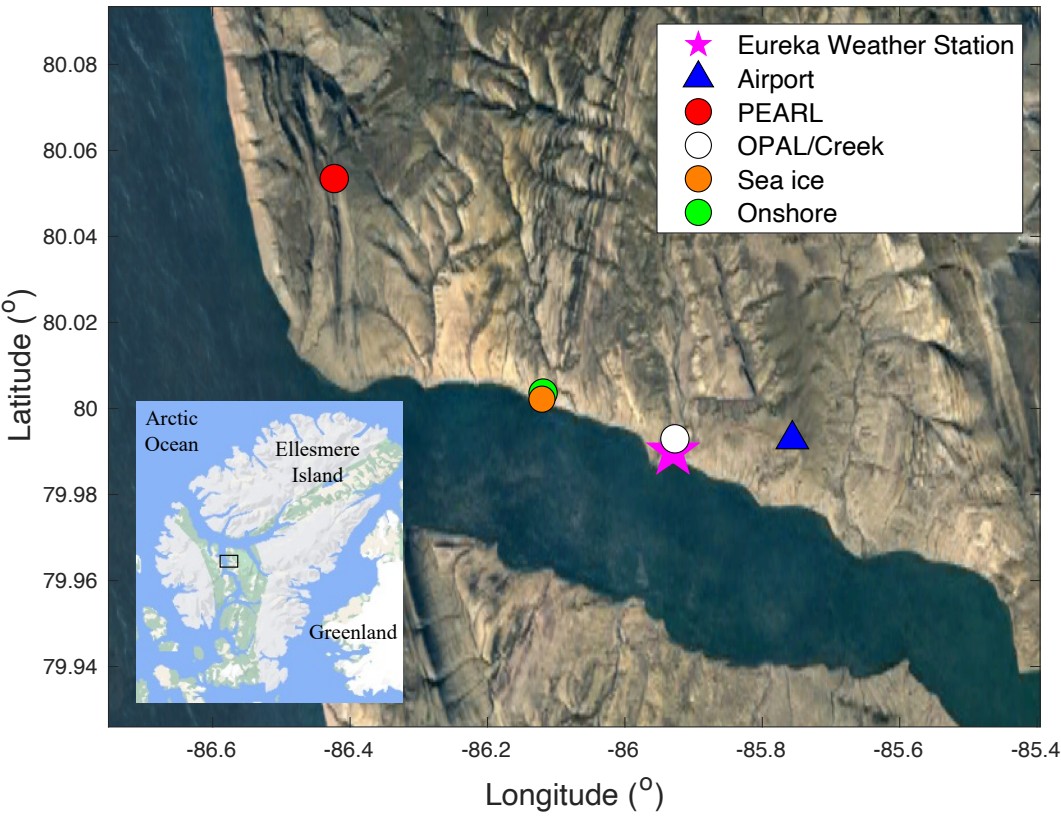

Figure 1. Local map with location of snow sampling sites marked by circles. The Eureka Weather Station (EWS) is marked by a star and the Eureka airport is marked by a triangle. The small inset box shows the location of the main map of Ellesmere Island, Canada. Image copyright: ©Google Earth/Google Maps.



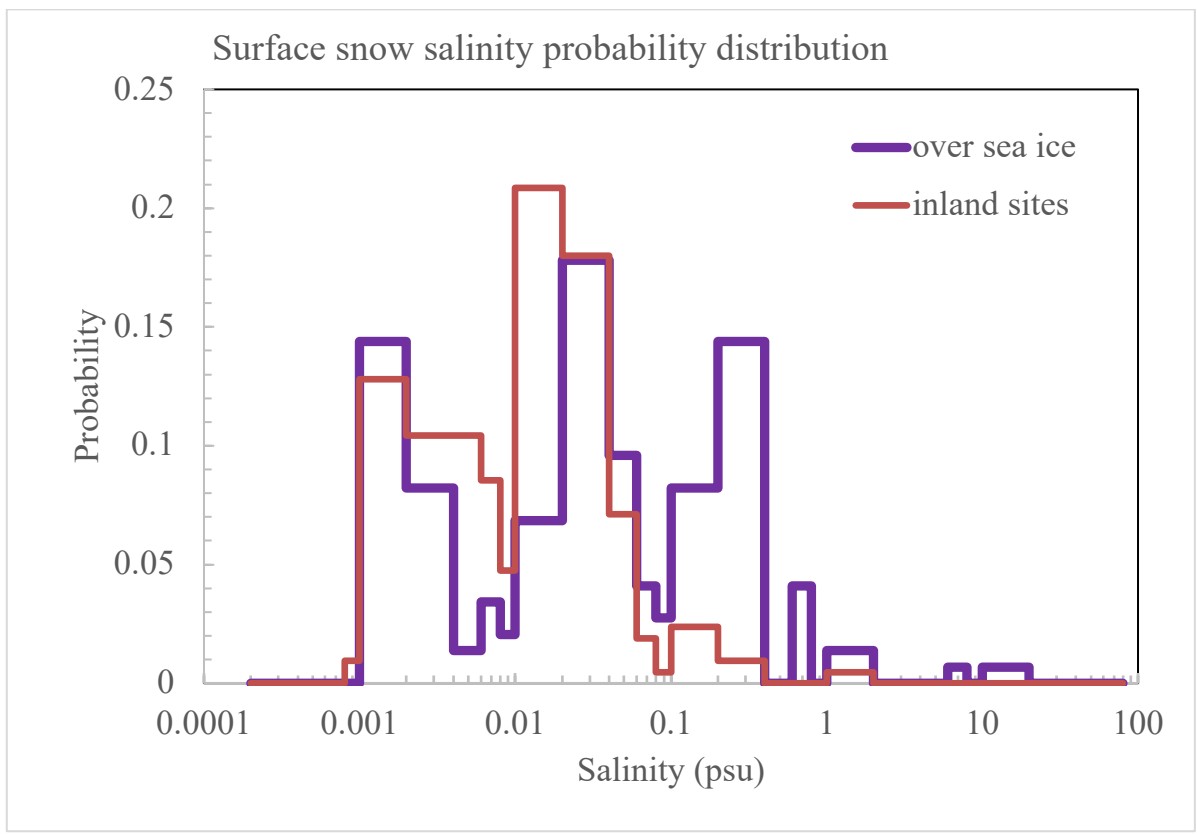

Figure 2. Eureka snow salinity probability distribution. The data include 2018 and 2019 snow sample measurements. The distribution over sea ice includes 146 snow samples, and the distribution at inland sites includes 211 snow samples (see Table 1).

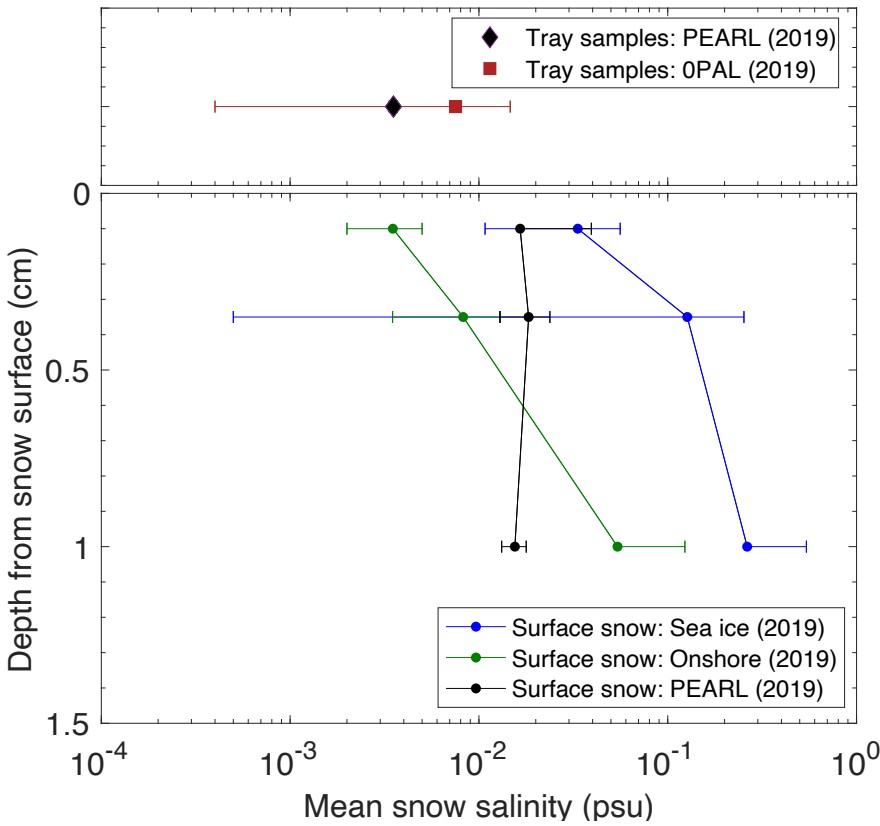

Figure 3. Mean snow salinity from the top 1.5 cm in three sub-layers: 0–0.2 cm, 0.2–0.5 cm, and 0.5–1.5 cm at the Sea ice, Onshore and PEARL sites (lower panel), and tray sample salinity at the 0PAL and PEARL sites (upper panel). The horizontal error bar represents one standard deviation. Note that tray samples at 0PAL were from a mounted tray outside the 0PAL building, approximately 1 m above the ground. Tray samples at PEARL were from a mounted tray (~1.5 m) on the roof of the PEARL Ridge Laboratory (~11 m above the ground).







Figure 4. Vertical profiles of 2019 snow ions [Na$^+$] (a), [Cl-] (b), [NO$_3^-$] (c), [Br$^-$] (d), non-sea-salt (nss)[Br$^-$] (e), nss[SO$_4^{2-}$] (f)
and enrichment factor of [Br$^-$] (g), [Cl$^-$] (h) and [SO$_4^{2-}$] (i) (see Section 3.2 for details).





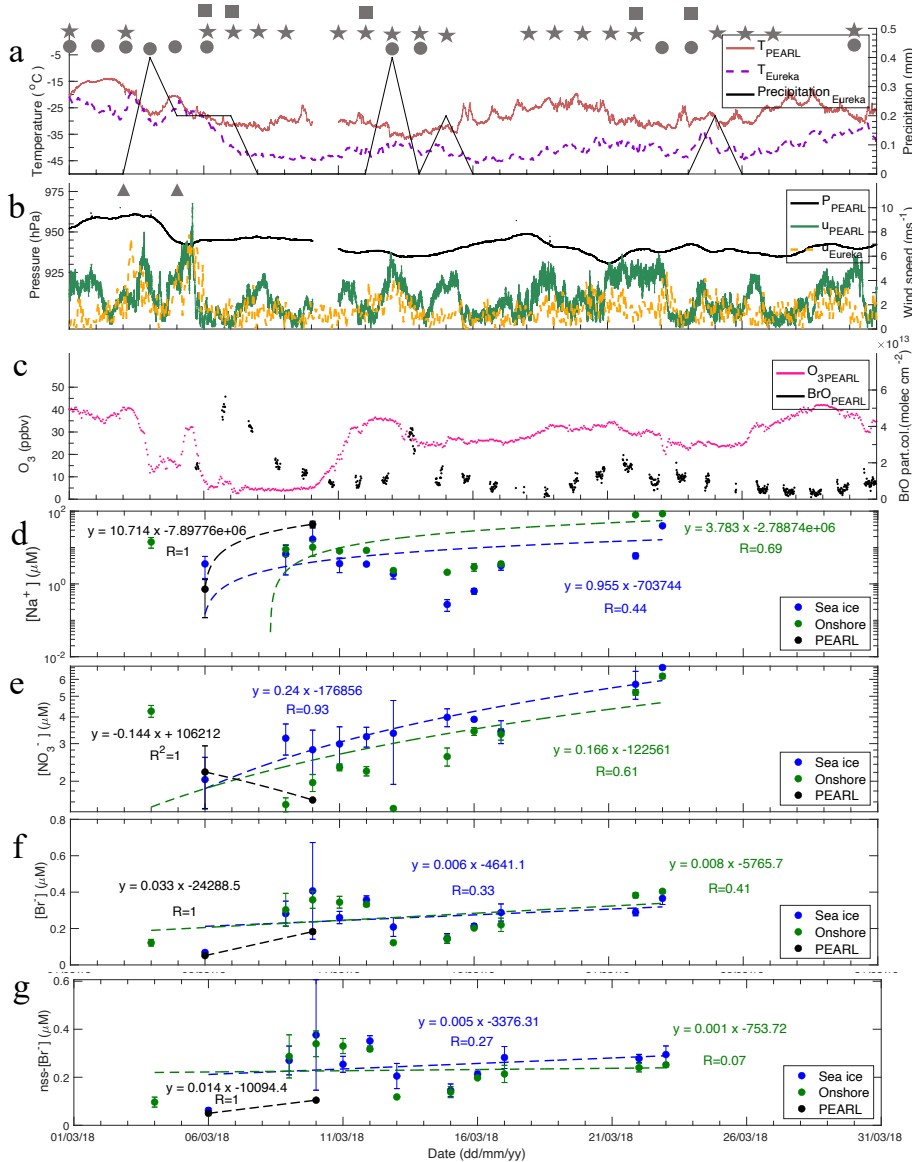

Figure 5. Time series of 2018 data. Air temperature at Eureka Weather Station (EWS) and PEARL and daily total precipitation
(≥0.2 mm) are shown in panel a. Local weather conditions are marked by symbols in panels a and b: squares representing fog
(>2 hours), stars representing ice crystal (>2 hours), circles representing trace precipitation (> 2 hours), and triangles
representing blowing snow (>2 hours). Atmospheric pressure at EWS and wind speeds at EWS and at PEARL Ridge
Laboratory are plotted in panel b; One-hourly surface ozone at 0PAL and MAX-DOAS BrO (0-4 km) partial columns from
the PEARL Ridge Laboratory are in panel c; Top 0.5 cm snow [Na$^+$] (d), [NO$_3^-$](e), [Br$^-$] (f) and nss[Br$^-$] (g) and corresponding
linear regression function against time and correlation coefficient R at each site are given. More statistical details of the linear
regressions in each panel are given in Table S4.



Figure 6. Same as Figure 5 but for 2019 snow time series. Noe that the meteorology data are only from the Eureka Weather Station and the ionic data are tray samples [Na$^+$] (d), [NO$_3^-$] (e), [Br$^-$] (f), and non-sea-salt (nss)[Br$^-$] (g). Table S4 gives more statistical details of the linear regressions in each panel. Local weather conditions are marked by symbols in panels a and b: squares representing fog, stars representing ice crystal, circles representing trace precipitation, and triangles representing blowing snow.





Figure 7. Time series of 2019 snow nitrate (a-c) and non-sea-salt bromide (d-f) in three sub-layers: 0–0.2 cm, 0.2–0.5 cm, and 0.5–1.5 cm at four sampling sites: Sea ice, Onshore, PEARL and 0PAL. Linear regression function against time and correlation coefficient R are given, see Table S4 for statistical details of the linear regressions.





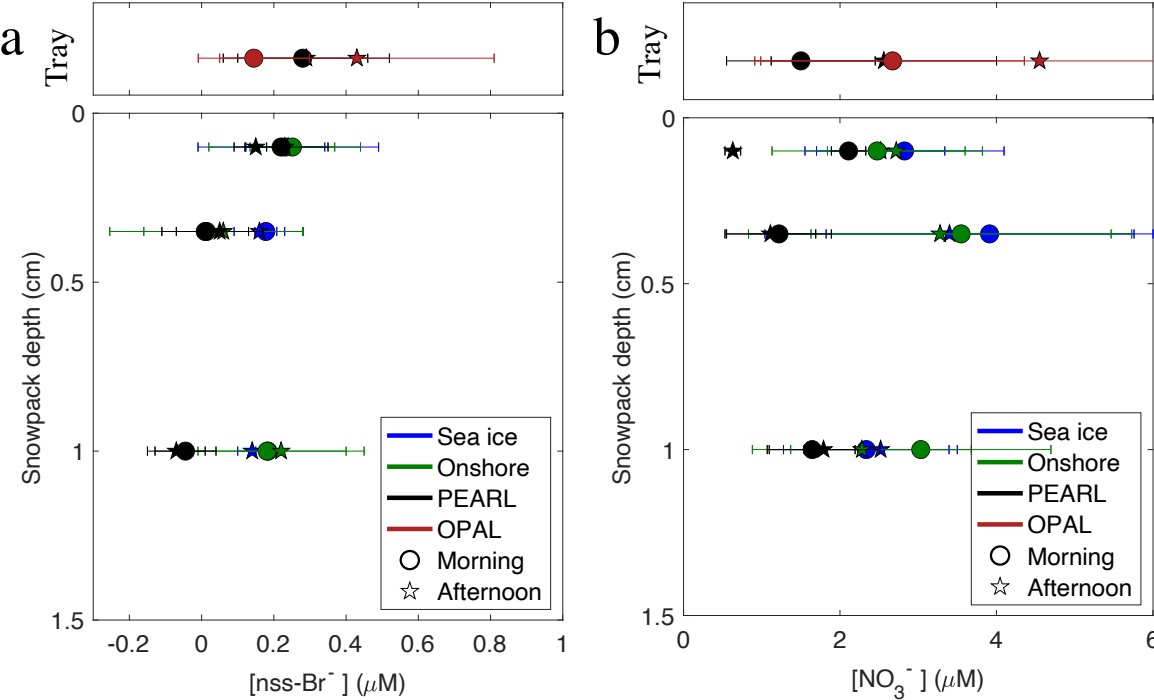

Figure 8. Morning and afternoon nss[Br⁻] (a) and [NO₃⁻] (b) from snow samples. The samples used in the analysis were collected mainly between March 3-16, 2019.




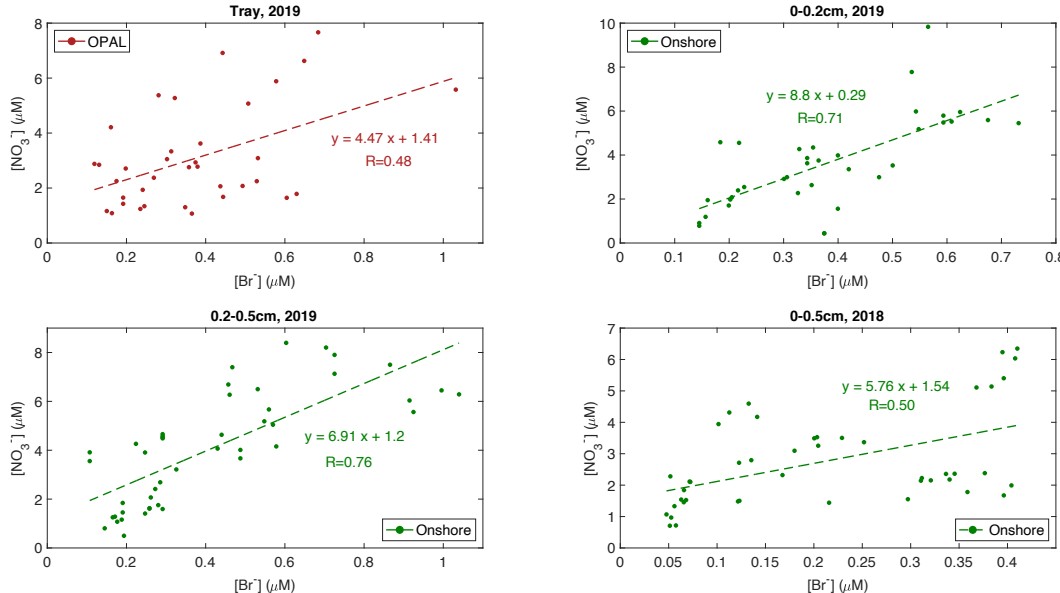

Figure 9. Scatter plot of surface snow nitrate versus bromide in (a) tray samples (2019), (b) 0–0.2 cm layer (2019), (c) 0.2–0.5 cm layer (2019), and (d) 0–0.5 cm layer snow (2018). Linear regressions and corresponding correlation coefficients R are given.