# Peer review of "Surface snow bromide and nitrate at Eureka, Canada in early spring and implications for polar boundary layer chemistry"

_EGUsphere, 2023_

## Author Comment (AC1)

**Reply to reviewer #1's comments**

We thank the reviewer for the helpful and constructive suggestions regarding how to improve the manuscript. The manuscript was revised carefully after considering the two reviewers' comments. Our responses are provided below in dark blue (and quoted texts added to the revision in italics), with the reviewer's comments included in black.

**Reviewer #1:**

Yang et al report observations of snow concentrations of ions along with surface atmosphere ozone and BrO concentrations at an Arctic location in the early spring of 2018 and 2019. To my knowledge, these are the first depth-dependent observations of snow bromide, which may be useful for understanding processes determining reactive bromine emissions from snow. They use time and depth-dependent measurements of bromide and nitrate to calculate the net deposition flux during the observational time period. They find that nitrate and bromide in snow are correlated and suggest that they are linked to one another through the formation and hydrolysis of bromine nitrate. They also find that deposition is confined to the surface skin layer.

This paper is very difficult to read, especially the long results section. There are a lot of details and numbers and it is presented in a way that makes it very difficult to discern the big picture. It reads like a first draft. The paragraph starting on line 424 is a particularly good example of this. There are more numbers than words in this paragraph and it is not readable. In general, the paper needs some reorganization and needs to be presented in a more succinct and readable manner. It often reads as a list of disconnected observations.

Response: We have revised the manuscript accordingly. In particular, we rewrote the paragraph which contains old lines 424-430.

*"Surface snow [Br⁻] and nss[Br⁻] show a very similar increasing trend (Figure 6(f) verse 6(g)), this is due to the large bromine enrichment factor or weak sea water impact. The 2019 tray sample nss[Br⁻] slope at 0PAL is $0.023\pm0.006$ $\mu M$ $d^{-1}$ (R=0.64, p<0.001, N=24 ), which is very close to the first layer slope values (Figure 7(d)) at Onshore and at Sea ice where the trends are statistic significant (p<0.02, Table S5). In the second snow layer, their slope values (Figure 7(e)) are still positive but with large p values (0.13-0.18). Tray sample nss[Br⁻] slopes at PEARL is $0.013\pm0.006$ $\mu M$ $d^{-1}$ (R=0.56, p=0.04, N=14)) which is smaller than that at 0PAL. In the first and second snow layer at PEARL, slope values are positive and statistic significant (Table S5)."*

A large portion of the results section focuses on salinity, but in the end, it is not clear what they learned from it as the results section is difficult to read and there is no follow-up on the salinity observations in the discussion or conclusions section. It is also unclear how an iceberg will impact snow salinity on sea ice and land.

Response: We added a few sentences in the text to address the link between snow salinity and snow ions observed. For example, in section 3.2:

*"Like snow salinity results, major snow ions on sea ice also have a large perturbation from 2018 to 2019. For example, the 2018 column mean snow sodium on sea ice (Tables S2 and S3) is 3-4 times of the 2019 column mean, which is consistent with the relatively low snow*

*salinity observed in 2019 due to the presence of a large iceberg grounded in the valley. In 2018, the column mean (1.5–20 cm) bromide on sea ice is 10.74±8.52 μM (N=80) (Table S2), while in 2019, it is only 6.47±5.36 μM (N=66) (Table S3). The lower 2019 snow bromide on sea ice is likely attributed to the freshwater dilution by the iceberg. However, they are much smaller than mean 30.6 μM on thick first year ice (FYI) and 92.5μM on thin FYI at Barrow, Alaska (Krnavek et al., 2012)."*

In section 5 of conclusion, we added a paragraph below to highlight the impact of the iceberg on snow salinity:

*"Skin layer snow salinity at the inland site has a double-peak distribution, with the first peak at 0.001-0.002 psu corresponding to the precipitation effect, and the second peak at 0.01– 0.04 psu is likely due to the salt accumulation effect. Snow salinity on sea ice has a triple-peak distribution, and the third peak at 0.2–0.4 psu is clearly related to the sea water effect due to the upward migration of brine on sea ice. The presence of an iceberg in the valley could significantly dilute ice and column snow salinity as observed in 2019 samples."*

Abstract line 26: missing a unit after 0.024.

Response: Done

Methods: State the eluants used for the IC measurements.

Response: The eluents for ion chromatography were generated with a Dionex hydroxide eluent generation cartridge (EGC) for anion analyses and a Dionex methanesulfonic acid EGC for cation analyses.

Line 342: I think you mean to say that the "concentrations" are larger, not the "profiles".

Response: You are right. corrected.

Line 419: What is a near zero increasing trend? Does this mean that the increasing trend is not statistically different from zero?

Response: Yes, the sentence has been rewritten to "the increasing trend is not statistically different from zero in PEARL tray samples and in the first layer."

Section 3.5: I think these calculations represent a net deposition flux (deposition minus emissions) and this should be explicitly stated.

Response: This point has been clearly addressed in the manuscript, e.g. in section 3.6, we stated *"where Flux is mean net deposition flux (deposition minus emission)"*.

---

## Author Comment (AC2)

**Reply to reviewer #2's comments**

      We thank the reviewer for the helpful and constructive suggestions regarding how to improve the manuscript. The manuscript was revised carefully after considering the two reviewers' comments. Our responses are provided below in dark blue (and quoted texts added to the revision in italics), with the reviewer's comments included in black.

**Reviewer #2:**

I find this manuscript confusing and the conclusions over-reaching the actual findings. The authors appear to see a net deposition flux of Br- to snowpack in one year (2019) and a much smaller one in the prior year (2018). The trend of net bromide deposition in 2019 appears to be out of error estimates. Beyond these solid findings, I think that the manuscript does not make clear arguments for how the net deposition flux of Br- is relevant to snowpack emission. Multiple other studies have shown that illuminated snowpack emits reactive halogens, and we know that reactive halogen eventually convert back to halides and deposit to snowpack. If these two processes (emission and deposition) are in balance, then the net deposition to snowpack can be near zero. Their finding of only small net deposition is not in conflict with other studies showing snowpack is a source of reactive bromine, and their finding of small net deposition does not preclude snowpack playing a significant role in reactive halogen production.

Response: We thank you for the constructive comments. To answer your questions raised, we performed further mass balance analysis for bromine in both surface snow and lower troposphere (<4km). As you will find below (a new section 3.7 is added in the revised manuscript) the new analysis still strongly indicates that surface snowpack in Eureka is not a large source of reactive bromine in March. Therefore, our major conclusion made previously remains unchanged.

Error analysis:

Aspects reported in this manuscript seem to be smaller than detection limits. On line 203, it is stated that the limit of detection (LOD) for Br- is 0.2 micromolar. It appears from the slope analysis (main text says it is Table S4, but it appears to be Table S5) that there are enough data points and a long enough time window (about 5-20 March in 2019) to get the slopes to be statistically significant despite this error on Br-.

Response: This is very true; we carefully designed the experiment by collecting daily snow samples over a relatively long period (of 3-4 weeks) with an aim of detecting the possible accumulative change.

Mass balance considerations:

Considering the Br- LOD (0.2 micromole / L) in comparison to the atmospheric column of BrO is instructive. Say the top 1.5cm of snow had density 0.3 g cm^-3 (from Figure S2), then the LOD of Br- (0.2 micromole / L) would be equivalent to: 0.2e-6 mole / 1000 cm^-3 water * 0.3 cm^3 water / 1 cm^3 snow * 1.5cm snow * 6.022e23 molecule / mole = 5e13 molecule cm^-2, which is comparable to the larger BrO partial columns. This calculation just points out that errors on the Br- analysis greatly complicate the interpretation of these data with respect to bromine activation.

Response: You are right, the LOD for bromide analysis is relatively large, almost ten times the average daily change of bromide in surface snow (~0.024 micromolar). A major reason why we performed snow sampling over a period of 3-4 weeks is to derive the possible large long-term trend instead of the small short-term variation.

Tray comparison:

When considering the surface snow, if there was one-way deposition of a species, it would build up in the snowpack and have an increase as is detected for Br- in the top snow, let's say at the 0.024 micromolar / day rate, so 0.024 micromolar is gained every day. It is not clear in the text, but one might assume that the tray samples are swept clean at each (daily) sampling, so there would not be an integration of the Br- over a longer period, but the tray would only have 0.024 micromolar of Br- in it each day. That would seem to say that there should not be a slope of the tray samples, but only a small fixed amount each day.

On the other hand, if there was an increasing flux of atmospheric Br- over this period, which Figure S1 shows to be a period of greatly increasing UV intensity, it might lead to an increasing amount of Br- in the tray deposit samples. One would then expect that the snowpack would be gaining Br- with an accelerating rate (because more Br- is coming down according to the daily tray samples), but that is not observed, possibly because the snowpack is producing reactive halogens that reduce the concentration of Br- in the snow, and we should not consider the deposition to be "one way".

Response: In the revised version, we added one sentence: *"The trays were swept clean daily after sampling using a clean brush"*.

Since the tray at 0PAL was mounted at a height of ~1 m above the ground (at PEARL, it was ~11 m above the ground and ~1.5 m above the building roof), during the ~24 hours period, snow samples collected were unavoidably influenced by surface snow under stronger winds. In section 2.2 we already stated that *"In windy conditions, most of the samples collected by trays consist of blowing snow particles"*. For this reason, we do not treat tray samples completely different from the skin layer snow. Almost same slope values were observed in the first snow layer and in tray samples, see Figures 6&7.

To address your question regarding whether the deposition is "one-way" or "two-way", we added a new section 3.7 (below) for bromine mass balance analysis:

**"3.7 Bromine mass balance analysis**

[revised manuscript text omitted]

Morning / afternoon differences:

Lines 463-471 are not very clear to me. They say that "signals are not significant across all sampling sites". From the text, it appears that all of the mentioned differences between morning and afternoon are well below the mutual error of the morning and afternoon samples. For example, on line 464, it says that morning is 0.25+/-0.12 micromolar, and afternoon is 0.23+/-0.21 micromolar. From these error bars, I would say that these numbers are the same. If they want to state that they are different, they would need to give numbers of data points and do careful statistics. Similarly, looking at Figure 8, visually examining the points, it seems 3-4 points on this plot, and the error bars seem to overlap a lot, but I cannot tell which error bar goes with which. I think that the morning/afternoon difference needs a clearer plot and an error analysis to be convincing. This section then concludes by reporting: "Based on the above numbers, a mean daytime bromide loss rate of 0.027 micromolar at sea level was obtained." I don't see how they put these conflicting numbers, all seemingly below error bars to finally result in a number that is about a factor of 7 below their Br- detection limit. Overall, this photochemical difference would need a better explanation to be believable.

Response: Figure 8 is updated and shown below, together with a new Table S6 added (shown below) for updated statistical data for morning and afternoon samples. We agree with you that due to the large error bars it is impossible to detect any signal in statistic confidence of the morning/afternoon difference (for both bromide and nitrate). Therefore, we rewrote section 3.5 and avoided claiming that daytime photochemistry signals were detected. We also deleted the relevant discussion on this in section 4.

*New Figure 8. Figure 8. Morning and afternoon nss[Br⁻] and [NO₃⁻] from available snow samples collected between March 3-16, 2019. Note that only the mean values with p<0.1 (in Table S6) are shown and used for the morning-afternoon difference calculation.*

[Figure]

Table S6: Nitrate and bromide analysis between morning and afternoon samples (i.e., mean(μM), one standard deviation(μM), number of observations, and *p*-values) for Figure 8.

| | | | Morning | | | | Afternoon | | | |
|---|---|---|---|---|---|---|---|---|---|---|
| | | | Mean | STD | N | p-value | Mean | STD | N | p-value |
| [NO₃] (µM) | Tray | PEARL | 1.50 | 0.95 | 4 | 0.050 | 2.20 | 1.25 | 9 | 0.001 |
| | | OPAL | 1.54 | 0.69 | 5 | 0.007 | 3.27 | 1.80 | 8 | 0.001 |
| | 0-0.2cm | Sea ice | 2.83 | 1.27 | 14 | 1.48E-06 | 2.52 | 0.82 | 8 | 5.43E-05 |
| | | Onshore | 2.48 | 1.34 | 12 | 5.23E-05 | 2.72 | 0.88 | 8 | 5.26E-05 |
| | | PEARL | 1.67 | 0.77 | 4 | 0.023 | 0.63 | 0.10 | 2 | 0.072 |
| | 0.2-0.5cm | Sea ice | 3.91 | 2.09 | 14 | 9.27E-06 | 3.40 | 2.36 | 10 | 0.001 |
| | | Onshore | 3.55 | 1.92 | 14 | 1.05E-05 | 3.28 | 2.45 | 12 | 0.001 |
| | | PEARL | 1.18 | 0.60 | 11 | 7.29E-05 | 1.11 | 0.58 | 5 | 0.013 |
| | 0.5-1.5cm | Sea ice | 2.34 | 1.06 | 16 | 2.49E-07 | 2.79 | 0.97 | 10 | 7.90E-06 |
| | | Onshore | 3.03 | 1.66 | 13 | 2.63E-05 | 2.26 | 1.72 | 14 | 2.82E-04 |
| | | PEARL | 1.51 | 0.64 | 17 | 3.72E-08 | 1.84 | 0.74 | 5 | 0.005 |
| [nss-Br] (µM) | Tray | PEARL | 0.28 | 0.18 | 8 | 0.003 | 0.30 | 0.24 | 9 | 0.005 |
| | | OPAL | 0.04 | 0.09 | 6 | 0.340 | 0.22 | 0.13 | 10 | 4.55E-04 |
| | 0-0.2cm | Sea ice | 0.23 | 0.11 | 13 | 5.88E-06 | 0.24 | 0.25 | 8 | 0.029 |
| | | Onshore | 0.25 | 0.12 | 12 | 1.31E-05 | 0.23 | 0.21 | 8 | 0.019 |
| | | PEARL | 0.24 | 0.14 | 12 | 6.56E-05 | 0.15 | 0.03 | 4 | 0.001 |
| | 0.2-0.5cm | Sea ice | 0.18 | 0.03 | 10 | 1.73E-08 | 0.13 | 0.09 | 6 | 0.022 |
| | | Onshore | 0.01 | 0.27 | 14 | 0.852 | 0.06 | 0.22 | 11 | 0.404 |
| | | PEARL | 0.01 | 0.13 | 10 | 0.783 | 0.04 | 0.11 | 5 | 0.400 |
| | 0.5-1.5cm | Sea ice | 0.18 | 0.03 | 4 | 0.002 | NaN | NaN | NaN | NaN |
| | | Onshore | 0.18 | 0.22 | 16 | 0.005 | 0.22 | 0.26 | 12 | 0.013 |
| | | PEARL | -0.08 | 0.08 | 11 | 0.008 | -0.05 | 0.11 | 5 | 0.329 |

Tray samples mass balance problems:

On lines 472-479, it is discussed that trays appear to gain Br- and NO3- over the day. If you wanted to try to compare the trays to snowpack, one would need to consider the amount of snow in each reservoir to calculate the mass of Br- in the tray and then compare to the mass of Br- in say the top of the snow pack. If there is not a lot of water mass in the tray, then a

small addition of Br- (mass) could increase its concentration much more than it would affect the larger reservoir of snow pack.  If they want to try to make a mass balance consideration of snowpack bromide emission and uptake of atmospheric particles, they need to consider the sizes of the reservoirs.

Response: It is true that the amount of water mass could affect the concentration. In the revised version, we added a new sentence in section 3.5 to highlight this effect:

*"The enhancement of tray sample's concentrations is likely due to the small amount of snow water collected by trays; the small addition of bromide deposited could increase its concentration much more than it would affect the large reservoir of surface snow."*

Deposition fluxes:

Section 3.5 attempts to calculate the net deposition flux of bromide from the increase in bromide in the top layers of snow.  They get a deposition flux at sea level, but don't have an error bar on this number.  From the standard deviation of the slopes given in Table S5, this should be possible to be calculated.  The slope errors appear to be on the order of 0.009 micromolar Br- / day.  If I compare that to the surface snow slope of about 0.02 micromolar Br- / day, that would be a fractional error of 0.009 / 0.02 = 0.45 or 45% relative error.  Therefore, my ball park calculation would indicate that the deposition flux is (1.01 +/- 0.45) x 10^7 molecule cm^-2 s^-1.  It appears unlikely that PEARL's error would be very different, so I'm not at all convinced that the quoted "At PEARL, the integrated flux is 7.9 x 10^-6 molecule cm^-2 s^-1, which is ~20% lower than at sea level." is actually true outside of mutual errors.  They then go on to say that this proves that snowpack at sea level is not a large source of reactive bromine.

Response: In the revised manuscript section 3.6, we added error bars for both bromide and nitrate fluxes. *For nitrate, "According to the statistical analysis results shown in Table S5, we can work out a mean slope error of 0.066 at sea level and 0.019 $\mu M\ d^{-1}$ at PEARL. If we compare that to the average slopes derived of 0.28 $\mu M\ d^{-1}$ at sea level and -0.018 $\mu M\ d^{-1}$, we can work out relative errors of 37% at sea level and 95% at PEARL. Therefore, we have an integrated nitrate deposition flux of $(2.6\pm0.37)\times10^8$ molecules $cm^{-2}\ s^{-1}$ at sea level and $(-1.0\pm1.06)\times10^8$ molecules $cm^{-2}\ s^{-1}$ at PEARL. These results indicate that surface snow at sea level is a net sink of atmospheric nitrate, and at the hilltop it is a source of reactive nitrogen; however, the negative flux derived at PEARL has a large error bar, indicating the flux has a large uncertainty."*

*For bromide: "Similarly, from Table S5 we can derive a mean slope error of 0.0096 $\mu M\ d^{-1}$ at sea level and 0.0059 $\mu M\ d^{-1}$ at PEARL (for the top two layers). If we compare that to the average slope of 0.02 $\mu M\ d^{-1}$ at sea level and 0.015 $\mu M\ d^{-1}$ at PEARL, we have relative errors of 48% at sea level and 39% at PEARL. Therefore the integrated bromide flux is $(1.01\pm0.48)\times10^7$ molecules $cm^{-2}\ s^{-1}$ at sea level and $(0.79\pm0.31)\times10^7$ molecules $cm^{-2}\ s^{-1}$ at PEARL."*

The small bromide deposition flux difference (<20%) between sea level and the hilltop indicates that these two sites likely underwent very similar bromine influence. If snowpack in the boundary layer is a large direct source of reactive bromine as proposed, then an enhanced gas-phase bromine loading in boundary layer is expected and should result in an enhanced deposition flux of bromide to surface snow at sea level. However, this is seen in the data. For

this reason, we say that local snowpack at sea level is not likely a large source of reactive bromine.

In Section 3.5, discussing (net) deposition flux of Br-, they make the statement "Therefore, if local snowpack on sea ice in the fiord is a large source of reactive bromine, an enhanced deposition flux at sea level should be detected." Other studies have shown that snowpack produces reactive bromine, which of course depletes snowpack of Br-. Therefore, the snowpack at sea level would be expected to be losing Br- by snowpack photochemistry, which they even claim to observe. Some or all of this later re-deposits, and if the net cycle of snowpack production followed by deposition are in balance, then the net trend of Br- in the snowpack would be very small. They show a very small net deposition flux of Br- in the snowpack, which can be perfectly consistent with snowpack production of reactive Br that then does atmospheric chemistry and eventually is converted back to Br- and deposits back to the snowpack. If snowpack 50km offshore produced reactive bromine through snowpack photochemistry, some of that could transport to their study region in a few hours (at 5m/s wind, 50km is traversed in under 3 hours), and then deposit explaining their small net deposition flux.

Response: The transport of air masses from bromine rich area over sea ice to inland does change the mass balance in both the atmosphere and the snowpack. In the revised version section 4 we added a paragraph to discuss it issue:

*" It is reasonable to assume that a rough balance of bromine can be reached in both the atmosphere and the snowpack over the sea ice zone. However, once bromine rich air masses transport from sea ice into inland tundra areas, the bromine budget balance breaks down. In particular, air starts to lose gas-phase bromine and snow begins to gain extra bromide from the air, as we observed at Eureka. However, if quick photochemical equilibrium is reached in surface snow, then we should see a stable bromide concentration in snow. The same goes for gas-phase bromine in the air. However, the significant decrease in BrO partial column in the lower troposphere and the increase in surface snow bromide strongly indicate that they do not reach a photochemical equilibrium state in early spring. Moreover, the significant trends show that it is very likely that snow photochemical release of reactive bromine is a very weak process, and the emission flux must be much smaller than the derived removal flux for gas-phase bromine and the snow bromide deposition flux, which is around $1 \times 10^7$ molecules $cm^{-2}$ $s^{-1}$."*

Nitrate -- bromide relationship:

I don't understand what the sentence from line 521-523 means, and they say that the data is not shown. If they want to make some claim, they should show data for it.

Response: A new Figure S7 is added (see below) to the revised manuscript to demonstrate the relationship between surface snow sodium and bromide at Sea ice and Onshore sites. We added new words in section 3.8 to describe it:

*"Figure S7 shows that at the Onshore site surface snow sodium and bromide are not significantly correlated apart from in the third layer. At Sea ice, surface snow sodium and bromide are largely correlated but with $[Br^-]/[Na^+]$ ratios larger than the sea water ratio (~0.0065) indicating that surface snow gains bromide from the air at Eureka, which is generally in line with the finding at coastal Alaska (Simpson et al., 2005)."*

[Figure]

Figure S7: Relationship between surface snow bromide and sodium at Sea ice and Onshore sites: (a) 0-0.5 cm (2018); (b) 0-0.2 cm (2019); (c) 0.2-0.5 cm (2019); and (d) 0.5-1.5 cm (2019).

Similarly, the discussion made later in this section states "the ratio of [NO3-]/[Br-] ranges form 3.5-6.8, indicating that one molecule of bromide deposited to the surface is likely accompanied by 4-7 nitrate molecules." They don't take into account that only the net deposition is being measured in their studies. Given that they don't get at underlying emission and deposition, I don't understand how to make sense of this ratio in terms of gas-phase chemistry (R1 and R2).

Response: In the revised version section 1, we rewrote the relevant paragraph:

*"It is well-known that $BrO_X$ can directly react with $NO_X$ via the following reactions R1 and R2:*

$$BrO(g) + NO_2(g) \rightarrow BrONO_2(g) \tag{R1}$$

$$BrONO_2(g) + H_2O(aq) \rightarrow HNO_3(g) + HOBr(g) \tag{R2}$$

*The product HOBr in R2 can photolyze to reform Br atoms (R3) which then react with ozone to form BrO (R4) to further oxidise $NO_X$ in R1.*

$$HOBr(g) + h\nu \rightarrow Br(g) + OH(g) \tag{R3}$$

$$Br(g) + O_3(g) \rightarrow BrO(g) + O_2(g) \tag{R4}$$

*Therefore, the net reaction of R1-R4 is:*
$$NO_2(g) + O_3(g) + H_2O(aq) + h\nu \rightarrow HNO_3(g) + O_2(g) \tag{R5}.$$

*This means that under sunlight and in the presence of bromine, ozone and $NO_X$ molecules will be consumed effectively via chain reactions. Thus, the presence of $BrO_X$ may accelerate the conversion from NOx to nitrate and influence the atmospheric nitrogen budget."*

*In revised section 3.8, we added further discussion:*

*"In early spring, due to the small solar zenith angle, atmospheric OH is very low, and the dominant pathway of oxidising NOx to form nitrate is via the chain reactions R1-R4. From the net reaction in R5 we can see that without net consumption of bromine, NOx and ozone can be effectively consumed, which means more than one NOx molecule can be converted to nitrate per bromine atom. Figure 9 shows that the ratio of [NO$_3^-$]/[Br$^-$] ranges from 3.5–6.8, indicating that one molecule of bromide deposited to the surface is likely accompanied by 4–7 nitrate molecules, attributed to the fast recycling of gas-phase bromine species before they deposit to the surface snow."*

Two-way fluxes:

Literature has long supported a snowpack source of NOx from nitrate photochemistry. This will cause a flux out of the snowpack. NOx can also convert back to nitrate, which has a fast deposition velocity and will deposit back to snowpack. They don't measure the flux of nitrate being lost from the snowpack photochemically, but only the "net" flux that is the deposition minus the loss. Similarly, for Br-, they only measure the net deposition flux, not either production or loss individually. It is not at all clear that this work has truly quantified the daytime loss of Br- from snowpack, and even if they did measure the net loss during daytime, there could still be faster emission plus some deposition during the day that could make the snowpack production rate faster than their daytime snowpack Br- loss. I think that the discussion in lines 550-578 may be trying to do a calculation to split their net deposition into component true emission and true deposition fluxes, but I cannot follow what they are saying here. In addition to not being able to follow it, the whole discussion seems to be built upon the "daytime loss" of 0.027 micromolar, which had no error analysis and doesn't appear significant from Figure 8. Overall, I think that the discussion in this section is not clear enough that I can even diagnose if their reactive bromine emission flux is realistic or if the range listed is based upon realistic error estimates.

Response: Thank you again for the in-depth thought on this. As discussed previously, due to the large error bars we could not derive robust conclusion regarding the daytime bromide loss, thus relevant text and discussions are modified in the revised manuscript.

The new mass balance analysis (shown above) indicates that photochemical loss of snow bromide is much weaker than we thought, and the emission flux of reactive bromine should be much smaller than the deposition flux of bromine from the air and to the surface snow (which is about $1 \times 10^7$ molecules cm$^{-2}$ s$^{-1}$). Therefore, the deposition of bromide to surface snow is more likely a "one-way" flux, rather than a "two-way".

Overall:

I think that this manuscript would need major revisions with improved error analysis and clearer discussion of how the observed net deposition flux is split into emission and deposition fluxes to be acceptable.

Response: We rewrote the relevant parts according to your suggestions with added new data and analysis, and hopefully they make our points much clear and make sense.

---

## Author Response (AR3)

**We thank the reviewer for their in-depth thoughts and comments. As you will see below that we responded carefully to all your questions raised. In the text below, the reviewer's comments are shown in black, and our replies shown in blue and italic.**

This manuscript describes interesting measurements of snowpack composition including salinity, bromide, and nitrate. Net deposition trends are observed, as well as profiles of ionic species in the snowpack and small-length scale differences in deposition around the Eureka area. However, the authors continue to try to say their data show that "the release flux of reactive bromine from snow must be a weak process and smaller than the derived bromide deposition flux of ~1×10^7 molecules cm-2 s-1, which flux is smaller than previously estimated flux by a factor of more than an order of magnitude." I believe the arguments they make on this point are flawed, as described below. I agree with the prior reviews that this manuscript is difficult to read and doesn't present a clear story, also making it hard to understand some of their arguments. I would suggest they clarify their description of observations and unless they can make a valid argument that their data actually constrains short-timescale snowpack emissions, remove that point.

*Response: As you will see below that we carefully further examined our major conclusions by showing new evidence, however, we do agree with the reviewer that the method applied in this study could not fully resolve short-timescale snowpack emissions. Thus, the flux derived only represents an AVERAGE flux over a long-term period (~1 month).*

Snow sampling and small flux challenges:

Fundamentally, the low concentration of bromide in the surface snow along with variability in ion concentrations related to sampling different surface snowpack (your next day's snow sample will be to the side of the prior day so you don't dig a hole) means that it is difficult to quantify changes in snow composition over time. The authors attempt to make up for this inherent challenge by using a fairly long timeseries. This approach means that the NET long-term deposition trend is quantified. Prior reviews pointed out that analysis errors and variability make it hard to quantify the net deposition trend, and the response was that "...collecting daily snow samples over a relatively long period of 3-4 weeks with an aim of detecting the possible accumulative change." This clearly shows that the study authors understand that they only detect the "accumulative change" = NET deposition.

*Response: Yes, the measured flux is a NET deposition flux.*

Net deposition does not constrain short-term snowpack emissions:

Net deposition is measured, but there may be larger fluxes that are bidirectional (snowpack emission and deposition) occurring on shorter timescales. Therefore, the authors cannot say that the long-term trend in NET deposition constrains shorter-term fluxes to be small. If the shorter-term fluxes were unidirectional only (e.g., deposition only), then the net deposition can constrain the process (there is no emission in this presumed unidirectional case), but in the case of bromine, we know that snowpack can both emit reactive halogens and have halogens deposit to it. The short term (days to hours) variability in both BrO and snow Br- also are consistent with significant short-term fluxes.

*Response: Note that the small net deposition flux measured could be attributed to two different scenarios: (1) the emission and the deposition fluxes are both large and the net flux is a residual of the two fluxes; (2) the net flux is determined mainly by the deposition flux, and the emission flux is relatively small and thus can be ignored. Obviously, scenario 1 is what the reviewer suggested, while scenario 2 is what we assumed. Note, the measured short-term (days to hours) variability in both BrO and snow Br- does not necessarily lead to snowpack emissions, as other mechanisms such as blowing snow can result in a similar episodic perturbation.*

New "Bromine mass balance" approach (section 3.7):

The authors make an equation for the time trend in air column density, equation R7, which says dc_air / dt = P_air - c_air / tau_air. They go on to say: "However, from Figures 5(c) and 6(c), we see a significant decreasing tend of BrO partial column, indicating the input term P_air is much smaller than the loss term, c_air / tau_air."

This doesn't make mathematical sense. For equation 7's left side to be negative, P_air should be smaller than c_air / tau_air, but a large value of P_air can be allowed as long as c_air / tau_air is larger.

As an example, let's say that there is no trend in gas-phase BrO (steady state), then the left side of R7 is zero, which means that 0 = P_air - c_air / tau_air, which gives the steady-state result: P_air = c_air / tau_air. Let's take tau_air to be 1 day = 86400s and say BrO is 3e13 molecule cm^-2 (typical value from their plots), and they assume BrO is 0.3 of gas phase Br, so gas-phase c_air = 1e14 molecule cm^-2. Then the production rate is P_air = 1e14 molecule cm^-2 / 86400s = ~1e9 molecule cm^-2 s^-1. This emission flux is within the quoted measurement (in the literature) of "snowpack bromine emission, a direct gradient measurement of Br2 and BrCl above a patch of snowpack was made near Utqiaġvik, Alaska (Custard et al., 2017), who reported emission fluxes of 0.7–12 × 10^8 molecules cm−2 s−1." Instead, the authors decided to choose a lifetime of reactive bromine of 42 days in 2019, and due to this choice, they get a much smaller emission flux. This lifetime of reactive bromine seems unreasonably long given the episodic nature of reactive bromine events, and is discussed below.

If we simply look at the BrO timeseries in Figures 5c and 6c, we can see that BrO varies significantly during individual days. BrO doubles on some days, and there are many instances where there is a factor of two difference in BrO between one day and the next. If we interpret this variability as due to local fluxes, one would clearly accept that BrO lifetime can be on the order of a day, which would allow snowpack emission fluxes comparable to the measured result from Custard et al., 2017.

*Response: As mentioned above there are two different possible scenarios (1 & 2) that could result in very small net deposition fluxes. If the system is bi-directional, then the derived flux (representing an average flux) underestimates the large short-term fluxes; however, if the system is unidirectional, then the derived net flux represents well the actual emission flux occurring.*
*Note that we did not arbitrarily choose a long "lifetime", the number of 42 days was derived from an exponential fit to the 2019 spring (March to May, not March) BrO data (see figure below and relevant text, which was not deliberately discussed in our previous response. The derived long "lifetime" should not be treated as the actual lifetime for reactive bromine as obtained in a more isolated air parcel. This is because the timescale was derived from long-term seasonal BrO data, where the system is open and affected by many factors, thus the long "lifetime" most reflects a seasonal decay of atmospheric bromine. Therefore, we used the term "seasonal decay lifetime" to distinguish it from the actual lifetime of reactive bromine.*

Complications with their lifetime analysis:

This study was done at a time when the reactive halogen season was declining a bit on a ~month timeframe. This slight decline in net BrO is calculated to be a loss of c_air over time in their equation 7. They then go on to use the same BrO data with an exponential fit to calculate a "loss rate" of reactive bromine. Use of the same data on both sides of equation 7 appears circular.

*Response: We think the reviewer might have misunderstood the method used to derive these two parameters. The mathematical solution of equation R7 is an exponential function, which can be rewritten as a linear function as long as the timescale of Tau_air (or "lifetime", =42 days) is much larger than 1 day. Therefore, the loss rates derived directly from the linear fit and from the exponential fit to the BrO data should give very similar results. With an exponential fit, one can work out precisely the loss rate at any given time, but they are not a mean loss rate, to get a mean loss rate for a period we need to do integration and averaging. To avoid this process, we can directly apply a linear fit to the same dataset (as we did in this study). Therefore, we did not apply the same data on both sides of R7, we only applied the same BrO data to derive the two parameters.*

Note that this "loss rate" is a NET loss rate over many weeks, not a loss rate that is specific to shorter term processes. Let's for the sake of argument say that they had stopped their study six days earlier in 2019. It appears that the trend in BrO over time would now be increasing, and if you fitted it to an exponential, you would not have a loss rate, but a growth rate or possibly flat (zero slope, infinite lifetime). They take the exponential loss rate to be be indicative of the lifetime (tau_air) of reactive bromine, but now the loss rate would be very small and the "lifetime" of reactive bromine would be longer than the 42 days they calculate in 2019 -- now reactive bromine might live well into the summer or even over multiple years, which is counter to observations. Is is obvious that you cannot extract the lifetime of reactive bromine in the way they are trying to

do it here. Without a constraint on tau_air, they cannot use equation 7 to calculate the magnitudes of the two terms on the right side, and thus they cannot determine the snowpack production flux.

*Response: The reviewer did raise a key issue regarding how to precisely constrain lifetime (or tau_air) and loss rate from the field data. Following referee's suggestion (removing the last six days from the March BrO data), we did perform the experiment with the result shown below. It gave a Tau_air of 40 days with a small R=0.38. When the whole March data are considered, the significance increases but the uncertainty remains large. In this study, instead of using the March BrO data, we took the whole spring season (Math to May) BrO into consideration. As shown below in new Figure S6 (in the supplementary material), BrO has a decline trend from March to May, which trend is statistically significant (R> 0.6). Using the regression fits we derived the 42 days "lifetime" and loss rate of 3E11 molecules cm2 d-1. As mentioned above this long timescale "lifetime" is different from the actual short lifetime of reactive bromine obtained in an isolated airmass, it more likely represents a seasonal decay of atmospheric bromine species, affected by many other factors involving both physical and photochemical processes. The decline of BrO from early spring to late spring was also exhibited in other years (from 2016 to 2018) at Eureka as shown in Bognar et al. (2020), which demands further investigation.*

[Figure]

*This figure shows the 2019 March BrO column data, in which an exponential fit to the data (excluding the last six days) was used to derive lifetime, as suggested by the reviewer.*

[Figure]

*New Figure S6. 2019 BrO column data from March 5th to May 31st, in which the exponential fit and the linear fit are inserted. Note that Julian day (X-axis) was introduced in this figure for the regression fits.*

To show that equation 7 cannot be used in this manner, consider this hypothetical situation. Say there was a sealed test tube with some liquid water and vapor in it. We now raise and lower the temperature, which will cause water to evaporate (P_air) and raise c_air or condense and lower c_air. Over a long time (many warming and cooling cycles), if you fit the timeseries of c_air to a slope, dc_air/dt would be close to zero (much like their long-term trend in gaseous bromine is fairly flat). By their method, they would then fit the c_air over time to an exponential, and say that the lifetime of the vapor is long (because the timeseries is on average flat), so tau_air

is very long, and the second term (c_air / tau_air) will go to near zero. Equation 7 will then be dc_air/dt = ~0 = P_air - c_air/tau_air (term = ~0), so you find P_air = ~0. Their interpretation is that this system has no evaporation of water (P_air = ~0), while in fact water is evaporating and condensing with potentially large fluxes. The failure is that you cannot use the net flux to constrain faster bi-directional fluxes.

Realistically, over multi day periods, the weather changes, airmass origins change, the sun rises, temperature warms on average. These factors lead to wide variability in BrO as shown on their figures. Yet, they fit a long-term trend through the data and call that the lifetime of reactive bromine as if there were a constant loss rate for the full campaign, nearly a month. It is clear that faster than monthly processes are needed to describe reactive halogen chemistry and that long-term "accumulative changes" do not directly constrain faster underlying bi-directional fluxes.

*Response: Unfortunately, we could not fully agree with the referee on this topic. The derived near-zero evaporation flux from the suggested experiment is "reasonable" in our view, this is because the tube is a sealed system and there should not be a net evaporation flux as a long-term mean. Here we suggested another similar experiment by replacing liquid water and water vapour in the tube by mercury and mercury vapour. Since mercury, compared to water, is less sensitive to temperature change, there will be no such large short-term fluxes during the experiment period, thus a near zero evaporation flux and a large lifetime number will be derived, but they are completely correct. Therefore, the key issue is not in the method itself, the interpretation of the results relies on whether there are short-term bi-directional fluxes in the system.*
*We agree with the referee that the method used here could not resolve short-term flux (if there are such short-term fluxes), thus in the revised version, we used "average" flux in our statements. For instance, in section 4 on discussion, we stated: "the method applied could not resolve short-term (<1 day) fluxes, therefore the derived fluxes only represent an average flux over the campaign period (3-4 weeks)."*

Summary:

Overall, the central problem with this manuscript is that the authors do not accept the difference between a net deposition rate measured over a month-long period and faster bi-directional fluxes that are occurring (based upon prior literature reports of snowpack emissions). They cannot constrain fast fluxes that happen in bi-directional manners by a long-term deposition flux. The BrO measurements, which vary by factors of two day to day could clearly be consistent with large snowpack emission on one day followed by deposition back to the snow the next day. Alternatively, look at the snowpack Br- timeseries, which shows a lot of variability. It is clear that analysis errors and snow sampling variability can affect variability in measured Br-, but one were to believe that this variability were real, it would indicate large fluxes of reactive bromine out of the snow and re-deposition of Br- back to the snow.

Over the long-term, due to bi-directional fluxes, the net change in snowpack bromine could be small (as is observed), but a lot of chemistry could have happened on shorter timescales than their long-term net trends can capture. Effectively, the approach described in this manuscript is not equipped to put short-term constraints on snowpack emissions fluxes.

If the authors want to report snowpack composition, vertical profiles in pits, and net deposition fluxes over long periods of time, I can see the publication of a manuscript showing those results. However, I see no validity in the attempts they have presented to constrain short-term snowpack emissions fluxes. If the authors seek to maintain that point, I argue for rejection of the manuscript. In this set of comments, numerical examples derived from their figures were shown to be consistent with snowpack emissions fluxes measured by others and reasonable lifetimes for reactive bromine. The lifetimes and fluxes of these faster processes fit with variability observed in both atmospheric reactive bromine and snowpack bromide.

*Response: We accepted the reviewer's key point that our method could not resolve faster short-term bi-directional fluxes, however, we defended our major conclusion of the weak snowpack emission flux, though this flux only represents a mean flux over the period of ~1 month. In section 5 we stated: "Through the mass balance analysis we conclude that the average emission flux from snow over the campaign period should be less than the average bromide deposition flux of ~1×10^7 molecules cm-2 s-1, which is an order of magnitude smaller than previously measured emission flux of 0.7–12 × 10^8 molecules cm−2 s−1 (Custard et al., 2017). Note that the net mean fluxes observed do not completely rule out larger bidirectional fluxes over shorter time*

*scales."*